# A Tighter Bound for Reward Learning in Reinforcement Learning from Human Feedback

**Guoxi Chen**                                                                *chen_guoxi@seu.edu.cn*
*School of Automation*
*Southeast University*

**Xing Chen**                                                                *xingchen19@jlu.edu.cn*
*School of Artificial Intelligence*
*Jilin University*

**Bo An**                                                                *boan@ntu.edu.sg*
*College of Computing and Data Science*
*Nanyang Technological University*

**Ya Zhang**                                                                *yazhang@seu.edu.cn*
*School of Automation*
*Southeast University*

**Reviewed on OpenReview:** *https://openreview.net/forum?id=EyMoFzI3Oz&referrer*

## Abstract

As a key component of reinforcement learning from human feedback (RLHF), reward learning directly influences the final learned policy. Unfortunately, existing theoretical estimation error bounds in reward learning rely on the complexity of the reward function class, unattainable optimal parameters, or non-zero constants independent of sample size, leading to uncomputable bounds that are meaningless for reward function classes with unknown complexity. To address this issue, this paper presents an analysis of parameter estimation for reward learning in RLHF under general function approximation, without imposing restrictions on the complexity of the reward function class. A tighter bound is provided without non-zero terms independent of the sample size. The optimal parameters are eliminated by applying linear approximation around the learned parameters. Additionally, the relationship between the preference dataset and the learned parameters is further examined to demonstrate how to efficiently collect data based on the current learned parameters. Inspired by the theoretical results, a novel offline RLHF algorithm with parameter constraints is proposed, restricting parameters to the valid space defined by the dataset. Furthermore, an online RLHF algorithm is proposed to iteratively optimize parameter learning and improve data collection efficiency. This work provides a tighter bound than previous studies and offers theoretical guidance for online data collection under general function approximation.

## 1 Introduction

Reinforcement learning from human feedback (RLHF) has shown great empirical success in various fields, including large language models (Liu et al., 2023; Ouyang et al., 2022), recommender systems (Xue et al., 2023a;b), robot training (Kupcsik et al., 2018; Bıyık et al., 2022), and autonomous driving (Wu et al., 2024; Sun et al., 2024). The standard RLHF framework typically consists of learning a reward function from a preference dataset and subsequently applying deep reinforcement learning to optimize a policy based on the learned reward (Dong et al., 2024; Li et al., 2024; Wang et al., 2023; Nika et al., 2024). As a key

capability, the reward function plays an essential role in the success of RLHF (Kausik et al., 2024; Zhou et al., 2024). Since the reward function is typically parameterized by a neural network and its learning involves adjusting the network's parameters, parameter estimation error becomes critically important in measuring the performance of reward learning.

Parameter estimation errors in reward learning are inevitable (Gao et al., 2023; Pan et al., 2024; Coste et al., 2023). In practice, it is necessary to establish an error bound to determine how the estimation error scales with sample size. This error bound also provides valuable insights for optimizing data collection strategies. Furthermore, in an ideal setting, it is intuitively expected that the error bound provides guidance for identifying an infinite dataset, such that the estimation error converges to zero. However, existing error bounds fail to fulfill this expectation.

Several studies (Zhan et al., 2023b; Zhu et al., 2023; Xiong et al., 2024; Du et al., 2024; Ge et al., 2024) have explored sample complexity, showing that the error bound converges to zero as the dataset size approaches infinity when the reward function is restricted to the class of linear functions. Zhu et al. (2023) extended the linear analysis to the nonlinear case. In their results, even with an infinite number of samples and all conditions satisfied, the learning error does not necessarily converge to zero. A few studies (Zhan et al., 2023a; Ji et al., 2024) focused on providing convergence guarantees for general function approximations. Their error bounds rely on the complexity constraints of the reward function class, unattainable optimal parameters, the supremum operator and the expectation operator. These factors make it difficult to compute error bounds and use them to support the design of online data collection strategies.

Inspired by the weight fine-tuning process in large language models (LLM), we compress the parameter solution space to achieve a tighter bound for parameter estimation in reward learning under general function approximation. We also establish the theoretical relationship between the dataset properties, the optimal parameters, and the learned parameters. The main contributions of this paper are summarized as follows.

(1) We propose an estimation error bound in the form $\widetilde{\mathcal{O}}(\frac{C(\text{data},\theta^*)\sqrt{d+\log\frac{1}{\delta}}}{\sqrt{n}})$, where $C(\text{data},\theta^*)$ is a constant dependent on the dataset and the optimal parameters $\theta^*$ (see Theorem 1 and Table 1 for details). This bound is tighter than that of Zhu et al. (2023), and provides a theoretical foundation for parameter optimization and online data collection to ensure the estimation error converges to 0 as the sample size $n \to \infty$.

(2) To avoid dependence on unavailable optimal parameters, we perform a Taylor expansion around the learned parameters, replacing the dependence on the optimal parameters with the learned parameters. We further derive an estimation error bound in the form $\widetilde{\mathcal{O}}(\frac{C(\text{data},\hat{\theta})\sqrt{d+\log\frac{1}{\delta}}}{\sqrt{n}})$, where $C(\text{data},\hat{\theta})$ is a constant dependent on the dataset and the learned parameters $\hat{\theta}$ (see Theorem 2 for details).

(3) Inspired by the theoretical results, we propose a parameter-constrained offline RLHF algorithm that is designed to ensure parameter learning occurs within the region effectively covered by the offline dataset. Furthermore, we develop a novel online RLHF algorithm. Utilizing iterative processes, we design a bi-level optimization framework to optimize parameter learning and improve data collection efficiency.

The proposed theoretical results can be applied to more general reward learning problems and offer theoretical guidance for online data collection.

## 2 Related Work

This section begins by discussing learning from pairwise comparisons and then explores reward learning in both linear and general function classes.

**Learning from pairwise comparison.** Learning from pairwise comparisons has been widely studied in the literature. Rajkumar & Agarwal (2014) explored the statistical convergence properties of some rank aggregation algorithms within a natural statistical framework. Shah et al. (2016) proposed the sharp upper

| | The proposed bound | Existing bound (Zhu et al., 2023) |
|---|---|---|
| Linear case (i.e., $b_2 = 0$) | $\frac{1}{\zeta(1-\frac{\lambda}{\lambda+\underline{\alpha}})}\sqrt{\frac{d+2\sqrt{d\log\frac{2}{\delta}}+2\log\frac{2}{\delta}}{n}}$ | $C\cdot\sqrt{\frac{d+\log(1/\delta)}{\gamma^2 n}+\lambda B^2}$ |
| Nonlinear case (i.e., $b_2 \neq 0$) | $\frac{a_1-\sqrt{a_1^2-4*\sqrt{\frac{d+2\sqrt{d\log\frac{2}{\delta}}+2\log\frac{2}{\delta}}{n}}*a_2}}{2a_2}$ | $\alpha_1\cdot\sqrt{\frac{d+\log(1/\delta)}{\alpha_2^2 n}+\left(\lambda+\frac{b_2}{\alpha_2}+b_1 b_2 B\right)B^2}$ |

Table 1: Comparison of the proposed bounds with previous studies.

and lower bounds on the optimal error in estimating the Bradley-Terry-Luce (BTL) models. Recently, several studies have started exploring this issue within the framework of RLHF. Unlike earlier research, which aimed to minimize the probability of disagreement in random pairwise comparisons, studies in RLHF focus on identifying the optimal strategy from a preference dataset. Both Skalse et al. (2023) and Li et al. (2024) highlighted the ambiguity in reward function learning within the context of pairwise performance comparisons, emphasizing the critical need for convergence conditions for reward learning.

**Learning within the linear function class.** Existing research on reward function learning has achieved good results in the linear function class. Xiong et al. (2024) studied the reverse-KL regularized contextual bandit for RLHF, and proposed efficient algorithms with finite-sample theoretical guarantees. Zhan et al. (2023b) proposed a strategy to collect preference samples and provided an error bound for the parameter estimation. Zhu et al. (2023) analyzed the convergence conditions of the maximum likelihood estimator (MLE) and its corresponding error bound for the linear reward function class. This paper extends the results of Zhu et al. (2023) and provides a tighter bound for general function approximations.

**Learning within the general function class.** The research most similar to our objective is that of Zhan et al. (2023a). Under the pessimistic learning assumption, they provided a bound on the error when the dataset has good coverage. However, this bound requires limiting the complexity of the reward function class. Subsequent studies also limited the complexity of the reward function class when analyzing the sample complexity of general function learning, such as Chang et al. (2024), Liu et al. (2024), Huang et al. (2024), and Ji et al. (2024). Overall, these studies establish the finite-sample suboptimality gap of the form $\widetilde{\mathcal{O}}(C_{\text{coverage}}^2\sqrt{\mathcal{N}_{\mathcal{R}}/N})$, where $\mathcal{N}_{\mathcal{R}}$ is the complexity of the reward model class $\mathcal{R}$, $N$ is the sample size, and $C_{\text{coverage}}^2$ is the concentrability coefficient to characterize the coverage of the preference dataset. However, the complexity constraints on the reward function classes restrict the types of reward functions that can be represented, resulting in the ground truth reward function falling outside of this class. Furthermore, $C_{\text{coverage}}^2$ involves the supremum operator, the expectation operator, and the unattainable optimal parameters, which makes the bound difficult to compute and ineffective in guiding the online data collection process (see Section 5.3 for details).

In this paper, we study the general function approximation learning problem from the perspective of parameter estimation. We propose a tighter bound than Zhu et al. (2023). Based on theoretical results, we naturally propose an offline RLHF algorithm to learn the parameters in the space well covered by the dataset. Furthermore, we propose an online RLHF iteration for parameter optimization and data collection.

# 3 Preliminaries

We begin by introducing the notation used in the paper. The Markov decision process is introduced in Subsection 3.1, and the data collection model of human preferences is detailed in Subsection 3.2.

*Notations:* $\mathbb{R}$ and $\mathbb{R}^n$ denote the set of real numbers and the $n$-dimensional Euclidean space, respectively. $\mathbb{E}[\cdot]$ describes the mathematical expectation. $\Delta(\mathcal{X})$ denotes the set of all probability distributions over the space $\mathcal{X}$. $\|\cdot\|_2$ is the 2-norm of a vector or the spectral norm when applied to a matrix. $\|x\|_\Sigma = \sqrt{x^{\mathrm{T}}\Sigma x}$ is the semi-norm of $x$, when $\Sigma$ is a positive-semidefinite matrix. $X \geq Y$ (especially, $X > Y$) means $X - Y$ is positive semi-definite (especially, positive definite). $I_d$ is the d-dimensional identity matrix. $\lambda_{\min}(\cdot)$ is the

smallest eigenvalue of a matrix. $\mathbb{P}(A)$ represents the probability that event $A$ will occur. $\mathrm{tr}(\cdot)$ is the trace of a matrix.

## 3.1 Markov Decision Processes

In this paper, we consider a finite-horizon MDP described by a tuple $\mathcal{M} = \{\mathcal{S}, \mathcal{A}, H, \{\mathcal{P}_h\}_{h=1}^H, \rho, \mathcal{R}\}$, where $\mathcal{S}$ is a state space, $\mathcal{A}$ is an action space, $H$ is the horizon length, $\mathcal{P}_h : \mathcal{S} \times \mathcal{A} \to \Delta(\mathcal{S})$ is the state transition at each step $h$, $\rho$ is the initial state distribution, and $\mathcal{R}$ is the reward space. Following Zhan et al. (2023a), we consider the reward function $\mathcal{R} : \mathcal{T} \to \mathbb{R}$ to include the entire trajectory, which is more general than the single-step reward, where $\mathcal{T} := (\mathcal{S} \times \mathcal{A})^H$ is the set of all possible trajectories. The reward defined on the state-action pair can also be viewed as a special case of $H = 1$.

At each step $h$, the agent executes action $a$ from state $s$ and transits to the next state $s'$ with probability $P_h(s'|s, a) \in \mathcal{P}_h$. At step $H$, the agent completes the entire trajectory $\tau \in \mathcal{T}$ and receives a trajectory-based reward $r(\tau) \in \mathcal{R}$.

A policy $\pi : \mathcal{S} \to \Delta(\mathcal{A})$ determines the probability distribution over actions, conditioned on the current state. After determining the reward function $r$, the value function $V_r^\pi : \mathcal{S} \to \mathbb{R}$ for a policy $\pi$ is defined as the expected reward, given by $r$ starting at state $s$ and following policy $\pi$. Mathematically, for any $s \in \mathcal{S}$, $V_r^\pi(s) := \mathbb{E}_{\tau \sim \pi(s)}[r(\tau)]$, where the expectation is taken over the state transition and policy-induced action distribution. The Q-function $Q_r^\pi : \mathcal{S} \to \mathbb{R}$ for a policy $\pi$ is defined analogously, i.e., $Q_r^\pi(s, a) := \mathbb{E}_{\tau \sim (\pi(s), a)}[r(\tau)]$. Given a reward function $r$, the goal is to find a strategy $\pi$ that maximizes the objective $J_r(\pi)$, where

$$J_r(\pi) = \mathbb{E}_{s \sim \rho} \mathbb{E}_{\tau \sim \pi}[r(\tau)]. \tag{1}$$

The optimal policy $\pi^*$ is defined as the policy that maximizes $r*$, i.e., $\pi^* = \arg\max_\pi J_{r^*}(\pi)$.

In the RLHF setting, we aim to learn the ground truth reward function $r^*$ from human preference data, and obtain its estimate $\hat{r}$. The corresponding optimization algorithm then finds the optimal policy $\hat{\pi} = \arg\max_\pi J_{\hat{r}}(\pi)$, which, however, may be suboptimal with respect to the ground truth objective function $J_r^*(\pi)$. We define the gap as

$$G(\pi^*, \hat{\pi}) = J_{r^*}(\pi^*) - J_{r^*}(\hat{\pi}). \tag{2}$$

We will show in Theorem 3 that the gap $G(\pi^*, \hat{\pi})$ is influenced by the learning error of the reward. In Subsection 3.2, we will detail the data collection and learning model for the reward.

## 3.2 Human Feedback Model

We focus on the problem that both the trajectories and preference labels come from a pre-collected dataset $\mathcal{D} = \{\tau_1^i, \tau_2^i, p_i\}_{i=1}^n$, where $n$ is the sample size. For every sample $i$, $\tau_1^i$ and $\tau_2^i$ are two different trajectories, and $p_i \in [0, 1]$ represents human preference. The samples are assumed to be i.i.d. Assume that a human provides preference labels by comparing trajectories based on an unobservable reward function $r^*$. Considering randomness, the label $p_i$ is treated as a random variable that satisfies

$$\mathbb{E}[p_i] = \frac{e^{r^*(\tau_1^i)}}{e^{r^*(\tau_1^i)} + e^{r^*(\tau_2^i)}}. \tag{3}$$

It is a more general model. When $p_i$ is a Bernoulli variable with probability $\frac{e^{r^*(\tau_1^i)}}{e^{r^*(\tau_1^i)} + e^{r^*(\tau_2^i)}}$, equation (3) simplifies to the widely used Bradley-Terry-Luce (BTL) model (Bradley & Terry, 1952; Christiano et al., 2017; Zhu et al., 2023; Zhan et al., 2023a). Equation (3) also allows experts to provide continuous preference labels when the difference between trajectories is subtle. As analyzed in Subsection 3.1, the crucial factor of (2) is the error of reward function learning. Following Zhu et al. (2023), we study this problem from the perspective of parameter estimation. Throughout the paper, we make the following assumption on the parameterization of the reward.

**Assumption 1.** *The ground truth reward $r^*$ can be parameterized by $\theta^* \in \mathbb{R}^d$.*

This assumption is common (Zhu et al., 2023; Zhan et al., 2023b), and we always use parameterized neural networks to learn reward functions, which essentially defaults to this assumption. In the subsequent analysis, we use $r_{\theta^*}$ to replace $r^*$ and $r_{\hat{\theta}}$ for $\hat{r}$, to highlight our objective of learning the parameters $\theta^*$. There have been many studies discussing the relationship between the reward learning error bound and the sample size. Zhu et al. (2023) and Zhan et al. (2023b) studied the case where the reward function is linear, and they proposed the error bound from the perspective of parameter convergence. As for nonlinear reward functions, Zhan et al. (2023b) proposed a loose bound on parameter convergence, which does not tend to 0 as the sample size increases. From another perspective, some studies have focused on the identification capabilities and the complexity of the reward function class. Under pessimistic learning conditions, Zhan et al. (2023a) provide an error bound for learning nonlinear reward functions. However, this bound imposes limitations on the cardinality and expressive capacity of the reward function class. These same limitations are also noted in Chang et al. (2024) and Liu et al. (2024).

In this paper, we analyze the estimation error of nonlinear learning from the perspective of parameter estimation, tightening the bound discussed in Zhan et al. (2023b). We show that, even without pessimistic learning conditions, the convergence of parameter learning can still be guaranteed if the preference data provides rich information in the gradient sense (see Theorem 1 for details). We will provide an example to show that this condition is intuitive. Without this condition being satisfied, no performance guarantees can be obtained even if the reward function follows a linear structure.

## 4 Problem Formulation

This section defines the problem statement. The standard RLHF typically involves a two-stage procedure Song et al. (2024): first, learning a reward function from a preference dataset using MLE; second, employing the learned reward function with standard reinforcement learning algorithms (such as PPO) to derive the optimal policy with respect to the learned reward function, as summarized in Algorithm 1.

---

**Algorithm 1** Standard RLHF

---

**Input**: offline dataset $\mathcal{D}$.
1. **MLE**: compute $r_{\hat{\theta}}$ by solving (11) and (12).
2. **Standard RL**: find $\hat{\pi} = \arg\max J_{\hat{r}}(\pi)$.

---

To approximate the ground truth reward $r^*$, according to Assumption 1, we define a parameterized function class $\mathcal{R} = \{r_\theta | \theta \in \Theta\}$, such as neural networks or other parametric models, to approximate the true ground truth reward. The parameter space is defined as

$$\Theta = \{\theta \in \mathbb{R}^d \mid \|\theta\| \leq B\}, \tag{4}$$

where $B$ is a positive constant to measure the bound of the parameter space. Similar to Ahmadian et al. (2024), the reward learning stage aims to find parameters $\theta \in \Theta$ that minimize the negative log-likelihood $L(\theta, \mathcal{D})$:

$$L(\theta, \mathcal{D}) = -\frac{1}{n}\left[\sum_{i=1}^{n} p_i \log \frac{e^{r_\theta(\tau_1^i)}}{e^{r_\theta(\tau_1^i)} + e^{r_\theta(\tau_2^i)}}\right] = -\frac{1}{n}\left[\sum_{i=1}^{n} p_i \log \frac{1}{1+e^{-\eta_\theta^i}} + (1-p_i)\log\frac{1}{1+e^{\eta_\theta^i}}\right], \tag{5}$$

where $\eta_\theta^i(\tau_1^i, \tau_2^i) = r_\theta(\tau_1^i) - r_\theta(\tau_2^i) \in \mathbb{R}$.

For convenience, we use shorthand $\eta_\theta^i := \eta_\theta^i(\tau_1^i, \tau_2^i)$, and all functions in subsequent sections will omit dependency on $\mathcal{D}$, except for Subsection 6.2.

Then, we make the following assumptions about the reward function class $\mathcal{R}$. The Assumptions 2-4 constrain the variation in the reward function, ensuring that bounded parameter changes lead to bounded changes in reward values. Similar assumptions have also been mentioned in research on nonlinear function problems by Zhu et al. (2023). It is also common in the study of nonlinear functions (Malanowski et al., 2020; An et al., 2023).

**Assumption 2.** *The reward function $r_\theta \in \mathcal{R}$ is continuous with respect to $\theta \in \Theta$.*

**Assumption 3.** *For any $\theta \in \Theta$, $\tau \in \mathcal{T}$, the following inequalities hold:*

$$|r_\theta(\tau)| \leq b_0, \tag{6}$$

$$\|\nabla r_\theta(\tau)\|_2 \leq b_1, \tag{7}$$

$$\|\nabla^2 r_\theta(\tau)\|_2 \leq b_2, \tag{8}$$

*where $b_0$, $b_1$, and $b_2$ are some positive constants to restrict the $r_\theta \in \mathcal{R}$ from the value, gradient, and Hessian with respect to $\theta$, respectively.*

**Assumption 4.** *The second-order derivative of the reward function is bounded, and its gradient with respect to theta is also bounded, i.e., for any $\tau \in \mathcal{T}$, $1 \leq i, j \leq d$*

$$\left|\frac{\partial^2 r_\theta(\tau)}{\theta_i \theta_j}\right| \leq c_1, \tag{9}$$

$$\|\nabla \frac{\partial^2 r_\theta(\tau)}{\theta_i \theta_j}\|_2 \leq c_2, \tag{10}$$

*where $\theta_i$ is the $i-th$ parameter of $\theta$. $c_1$ and $c_2$ are some positive constants to restrict the second-order derivative of $r_\theta$ that satisfies the Lipschitz continuity condition.*

In the following probably approximately correct (PAC) analysis, we further assume that the parameter class is realizable.

**Assumption 5.** *(Realizability): $\theta^* \in \Theta$.*

Assumption 5 limits the analysis to the PAC learnability framework, where non-PAC learnable problems can only yield approximately optimal solutions within the solution space. This assumption is common in related research (Zhu et al., 2023; Zhan et al., 2023b; Wang et al., 2023; Liu et al., 2024).

Under the Assumptions 2-5 the MLE estimator attempts to solve the following optimization problem (Zhu et al., 2023; Zhan et al., 2023b).

$$\arg\min_\theta \quad L(\theta), \tag{11}$$

$$\text{s.t.} \quad \|\theta\| \in \Theta. \tag{12}$$

Now, we can clearly define the main focus. Define $\hat{\theta}$ as the last trained parameter obtained from problem (11)-(12), and the estimation error is $\Delta = \hat{\theta} - \theta^*$. In existing research, under similar assumptions, Zhu et al. (2023) establishes a bound for $\Delta$ in the form $\widetilde{\mathcal{O}}(\frac{C'\sqrt{d+\log\frac{1}{\delta}}}{\sqrt{n}} + C''b_2)$, where $C'$ and $C''$ are non-zero constants dependent on the dataset. The estimation error bound is expected to tend to 0 as $n$ increases if suitable samples are collected. However, the theoretical result falls short of this expectation if the reward function is non-linear (i.e., $b_2 \neq 0$). Some studies Zhan et al. (2023a) analyze the problem from the perspective of function expressiveness. However, these approaches inevitably require constraining the complexity of the reward function class, and the resulting bounds often involve an unattainable optimal function (see Section 6 for a detailed comparison). In this paper, we focus on providing tighter bounds for $\Delta$ than Zhu et al. (2023), and we further decouple $\Delta$ from its dependence on $\theta^*$, making it both computable and practical for guiding data collection.

## 5 Tighter Bounds for RLHF with General Function Approximation

In this section, we provide the main theoretical results, including tighter bounds for reward learning and tighter bounds for the suboptimal performance gap. In subsection 5.1, we tighten the bound on a key Hessian matrix to propose a tighter bound than that in Zhu et al. (2023) (Theorem 1). Similar to their result, this bound is related to the dataset properties and the optimal parameters $\theta^*$. To further eliminate dependence on the inaccessible optimal parameters, we perform a Taylor expansion around the learned parameters,

replacing dependence on the optimal parameters with that on the learned ones. This new error bound depends solely on the dataset properties and the learned parameters. In subsection 5.2, we establish the relationship between the gap $G(\pi^*, \hat{\pi})$ between the suboptimal strategy $\hat{\pi}$ and the optimal strategy $\pi^*$ and the reward learning error $\Delta$. This relationship is summarized in Theorems 3 and 4, which correspond to the results of Theorems 1 and 2, respectively. In subsection 5.3, we theoretically compare the estimated error bounds with those from existing research. We further explain why these bounds are tighter. Additionally, we provide explanations for the key parameter $\underline{\alpha}$ involved in the error bounds with an illustrative example.

## 5.1 The Tighter Bounds for Reward Learning

For convenience, define $\zeta = \min_{\theta,i} \frac{e^{\eta_\theta^i}}{\left(1+e^{\eta_\theta^i}\right)^2}$. Since $\eta_\theta^i$ is bounded, $\zeta$ must satisfy $0 < \zeta < 1$. A tighter bound for reward learning from the point of view of $\theta^*$ is summarized in Theorem 1.

**Theorem 1.** *Under the Assumption 2-5, if $\underline{\alpha} > 0$, for any $\lambda \geq 0$, with probability $1 - \delta$, (1) When $c_1 = 0$, we have*

$$\|\Delta\|_{\Sigma_D(\theta^*)+\lambda I_d} \leq \frac{a_0}{a_1}. \tag{13}$$

*(2) When $c_1 > 0$, $\exists n^*$, after $n > n^*$, we have*

$$\|\Delta\|_{\Sigma_D(\theta^*)+\lambda I_d} \leq \frac{a_1 - \sqrt{a_1^2 - 4*a_0*a_2}}{2a_2}, \tag{14}$$

*if $B < \frac{a_1+\sqrt{a_1^2-4*a_0*a_2}}{4a_2\sqrt{4b_1^2+\lambda}}$, where $a_2 = \underline{\beta}^3(8\zeta b_1 b_2 + b_1 c_1 d + 2c_2 d), a_1 = \zeta - \underline{\beta}^2(\zeta\lambda + dc_1\sqrt{\frac{8\log(2d^2/\delta)}{n}}), a_0 = \sqrt{\frac{d+2\sqrt{d\log\frac{2}{\delta}}+2\log\frac{2}{\delta}}{n}}, \underline{\beta} = \frac{1}{\sqrt{\lambda+\underline{\alpha}}}$, and $\underline{\alpha}$ is the minimum eigenvalue of $\Sigma_D(\theta^*)$, with $\Sigma_D(\theta^*) = \frac{1}{n}\sum_{i=1}^n \nabla\eta_{\theta^*}^i (\nabla\eta_{\theta^*}^i)^T$.*

Overall, the bounds depend on the parameters of the reward function class, i.e., $b_1$, $b_2$, $c_1$, $c_2$, $d$, the quality of the dataset, i.e., $\underline{\alpha}$, and the number of samples, i.e., $n$, while $a_2$ characterizes the part of the reward function class that captures nonlinearity. Comparing (13) and (14), both are derived from (15). Specifically, for the linear case where $a_2 = 0$, (15) can be further simplified to (13). This relationship is analogous to a quadratic equation with a second-order coefficient. More clearly, the linear case bound can be written as $\frac{1}{\zeta(1-\frac{\lambda}{\lambda+\underline{\alpha}})}\sqrt{\frac{d+2\sqrt{d\log\frac{2}{\delta}}+2\log\frac{2}{\delta}}{n}}$.

We sketch the proof of Theorem 1. The core goal of the proof is to construct the following quadratic inequality (15) that is only related to $\|\Delta\|_{\Sigma_D(\theta^*)}$.

$$-a_2\|\Delta\|_{\Sigma_D(\theta^*)+\lambda I_d}^2 + a_1\|\Delta\|_{\Sigma_D(\theta^*)+\lambda I_d} - a_0 \leq 0. \tag{15}$$

To achieve this goal, we use the PAC assumption (Assumption 5) to obtain the following inequality.

$$\|\Delta\|_{\Sigma_D(\theta^*)+\lambda I_d}\|\nabla l(\theta^*)\|_{(\Sigma_D(\theta^*)+\lambda I_d)^{-1}} \geq l(\theta^* + \Delta) - l(\theta^*) - \Delta^T\nabla l(\theta^*). \tag{16}$$

To express the right-hand side of equation (16) solely in terms of $|\Delta\|_{\Sigma_D(\theta^*)+\lambda I_d}$, we make use of a property satisfied by $\theta'$, which lies between $\theta^*$ and $\hat{\theta}$, rather than applying it to all $\theta$ as in Zhu et al. (2023). Then, at $\theta^*$, a Taylor expansion with a Lagrange remainder term is performed, and

$$\|\Delta\|_{\Sigma_D(\theta^*)+\lambda I_d}\|\nabla l(\theta^*)\|_{(\Sigma_D(\theta^*)+\lambda I_d)^{-1}} \geq$$
$$\zeta\|\Delta\|_{\Sigma_D(\theta^*)}^2 - \Delta^T(8\zeta b_1 b_2 + b_1 c_1 d + 2c_2 d)\|\Delta\|_2\Delta - dc_1\sqrt{\frac{8\log(2d^2/\delta)}{n}}\|\Delta\|_2^2. \tag{17}$$

can be further obtained. An important step in this process is tightening the bound on the following Hessian matrix $\frac{1}{n}\sum_{i=1}^n y_i\nabla^2\eta_{\theta'}^i$, where $y_i$ is a zero-mean random variable. We use a toy example in Appendix A.4

to illustrate that it may tend to 0 as $n$ increases, rather than being merely bounded by a constant. This is the key motivation to tighten the existing bound. Due to the correlation between $\theta'$ and $y_i$, we transform $\nabla^2 \eta^i_{\theta'}$ into a finite increment related to $\theta^* - \theta'$ according to Assumption 4 and a variable independent of $y_i$. This allows us to control the upper bound of the Hessian matrix at the element level, tightening the original upper bound $2b_2 I_d$ provided by Zhu et al. (2023). Furthermore, on the left-hand side of Equation (17), $\|\nabla l(\theta^*)\|_{(\Sigma_D(\theta^*)+\lambda I_d)^{-1}}$ is transformed into a lower bound that depends on $\sqrt{n}$, in a probabilistic sense. Since $\|\nabla l(\theta^*)\|_{(\Sigma_D(\theta^*)+\lambda I_d)^{-1}}$ only involves $\theta^*$, this step can follow a similar process to Zhu et al. (2023). Ultimately, this leads to a quadratic inequality in the form of Equation (15). The proof is detailed in Appendix C.1.

Parameters $a_2$, $a_1$, $a_0$, and $\underline{\alpha}$ are only related to the $\theta^*$ and dataset $\mathcal{D}$, so they are considered constants. $\Sigma_D(\theta^*)$ depends on the dataset and the ground truth parameter $\theta^*$. Thus, Theorem 1 shows that when the dataset satisfies some conditions, the estimation error tends to 0 as $n$ approaches infinity. It is a tighter bound than Zhu et al. (2023). However, this theorem is passive, as $\Sigma_D(\theta^*)$ is fixed, making Theorem 1 inapplicable to parameter learning and data collection. To overcome this limitation, we expand around $\hat{\theta}$ to associate the learned parameters with the properties of the dataset. A computable bound of the reward learning from the standpoint of $\hat{\theta}$ is shown in Theorem 2.

**Theorem 2.** *Under the Assumption 2-5, if $\underline{\alpha} > 0$, for any $\lambda \geq 0$, with probability $1 - \delta$, (1) When $c_1 = 0$, we have*

$$\|\Delta\|_{\Sigma_D(\hat{\theta})+\lambda I_d} \leq \frac{\hat{a}_0}{\hat{a}_1}. \tag{18}$$

*(2) When $c_1 > 0$, $\exists n^*$, after $n > n^*$, we have*

$$\|\Delta\|_{\Sigma_D(\hat{\theta})+\lambda I_d} \leq \frac{\hat{a}_1 - \sqrt{\hat{a}_1^2 - 4 * \hat{a}_0 * \hat{a}_2}}{2\hat{a}_2}, \tag{19}$$

*if $B < \frac{\hat{a}_1 + \sqrt{\hat{a}_1^2 - 4*\hat{a}_0*\hat{a}_2}}{4\hat{a}_2\sqrt{4b_1^2+\lambda}}$, where $\hat{a}_2 = \underline{\hat{\beta}}^3(8\zeta b_1 b_2 + b_1 c_1 d + 2c_2 d) \geq 0, \hat{a}_1 = \zeta - \underline{\hat{\beta}}^2(\zeta\lambda + dc_1\sqrt{\frac{8\log(2d^2/\delta)}{n}}), \hat{a}_0 = \underline{\hat{\beta}} b_1 \sqrt{\frac{8d\log(2d/\delta)}{n}}. \underline{\hat{\beta}} = \frac{1}{\sqrt{\lambda+\underline{\hat{\alpha}}}}$, and $\underline{\hat{\alpha}}$ is the minimum eigenvalue of $\Sigma_D(\hat{\theta})$, with $\Sigma_D(\hat{\theta}) = \frac{1}{n}\sum_{i=1}^{n}\nabla\eta^i_{\hat{\theta}}(\nabla\eta^i_{\hat{\theta}})^{\mathrm{T}}$.*

The sketch of the proof is similar to Theorem 1, except that the Taylor expansion is performed around $\theta'$. This causes $\|\nabla l(\theta^*)\|_{(\Sigma_D(\hat{\theta})+\lambda I_d)^{-1}}$ related to both $\hat{\theta}$ and $\theta^*$, which invalidates the proof strategy employed in Theorem 1. We solve this problem by bounding $\|\nabla l(\theta^*)\|$ by controlling the bounds and growth rate of its elements. Note that each element of $\nabla l(\theta^*)$ is a zero-mean random variable scaled by a constant. Therefore, the corresponding bound can be derived using the union bound. Then, we use the compatibility of norms to unify the resulting quadratic inequality. The proof is detailed in Appendix C.2. Parameters $\hat{a}_2$, $\hat{a}_1$, $\hat{a}_0$, $\underline{\hat{\beta}}$, and $\underline{\hat{\alpha}}$ offer a concise representation, highlighting the relationship between these parameters and $\hat{\theta}$. Theorem 2 establishes the relationship between the learned parameters $\hat{\theta}$ and the bound $\Delta$ on estimation error. This theorem suggests that it is possible to learn parameters within a finite space to ensure bounded estimation error. Moreover, it indicates that by continuously collecting data online, the coverage of the dataset can be optimized.

## 5.2 The Tighter Bounds for the Suboptimal Performance Gap

In this subsection, we address the second stage of standard reinforcement learning, which involves using reinforcement learning algorithms (such as TD3 and PPO) to optimize the policy based on the learned reward function. We summarize the performance gap between the policy induced by the learned reward function and the optimal policy in Theorem 3 and 4.

**Theorem 3.** *Under Assumption 2-5, the gap in performance between the policy induced by the learned reward function $r_{\hat{\theta}}$ and the ground truth optimal policy can be expressed as*

$$G(\pi^*, \hat{\pi}) \leq 2b_1\underline{\beta}\|\Delta\|_{\Sigma_D(\theta^*)+\lambda I_d}. \tag{20}$$

Recall that $\Delta = \hat{\theta} - \theta^*$ is the reward learning error of the parameters. The main idea of the proof involves converting the performance gap into the learning error of the reward function and obtaining the further result according to the Lipschitz continuity condition (Assumption 2). The detailed proof is provided in Appendix C.3. The bound of Theorem 3 is related to Theorem 1 from the perspective of $\theta^*$. Similarly, the result corresponding to Theorem 2 is summarized as Theorem 4.

**Theorem 4.** *Under Assumption 2-5, the gap in performance between the policy induced by the learned reward function $r_{\hat{\theta}}$ and the ground truth optimal policy can be expressed as*

$$G(\pi^*, \hat{\pi}) \leq 2b_1\underline{\hat{\beta}}\|\Delta\|_{\Sigma_D(\hat{\theta})+\lambda I_d}, \tag{21}$$

The proof sketch is similar to that of Theorem 3. The detailed proof is provided in Appendix C.4. In the next section, we will analyze in detail what these boundaries signify and compare them with existing results.

Theorems 3 and 4 reveal the relationship between reward function learning and the final suboptimal policy error, suggesting that the focus of theoretical analysis can be placed on reward function learning. Theorem 3 links the performance gap to the parameter estimation gap. Equation (20) implies that, with the same $1-\delta$ probability, the performance gap and the parameter estimation gap share the same upper bound, differing only by a constant. Theorem 4 has the same meaning as Theorem 3, except that it is linked to the parameter estimation error used in Theorem 2.

### 5.3 Insight into the Tighter Bounds

In Theorem 1, since $\underline{\alpha}$ is the minimum eigenvalue of the positive definite matrix $\Sigma_D(\theta^*)$, $\underline{\alpha} \geq 0$ always holds.

When $\underline{\alpha} > 0$, we compare the proposed bounds with existing studies. Table 1 shows that we achieved a tighter bound. In Zhu et al. (2023), for the linear case (i.e., $b_2 = 0$), when $\lambda \neq 0$, they achieve

$$\|\Delta\|_{\Sigma_D(\theta^*)+\lambda I} \leq C \cdot \sqrt{\frac{d + \log(1/\delta)}{\gamma^2 n} + \lambda B^2}, \tag{22}$$

where $C$ is a universal constant. The right side of the (22) does not tend to 0 as $n \to \infty$, if $\lambda > 0$. The condition $b_2 = 0$ implies $c_1 = 0$. As $n \to \infty$, $a_0$ vanishes at a rate of $O(1/\sqrt{n})$, while $a_1 = \frac{1}{\zeta(1-\frac{\lambda}{\lambda+\underline{\alpha}})}$ is a constant. Thus, $\frac{a_0}{a_1} \to 0$, if $\underline{\alpha} > 0$. Comparing (13) and (22), since the estimation error bound is a strictly monotonically decreasing function of $n$, there exists a value $\bar{N}$ such that for $n > \bar{N}$,

$$\|\Delta\|_{\Sigma_D(\theta^*)+\lambda I} \leq \frac{a_0}{a_1}$$

$$= \frac{1}{\zeta(1-\frac{\lambda}{\lambda+\underline{\alpha}})}\sqrt{\frac{d + 2\sqrt{d\log\frac{2}{\delta}} + 2\log\frac{2}{\delta}}{n}} \quad \text{(Our bound)}$$

$$\leq C \cdot \sqrt{\frac{d + \log(1/\delta)}{\gamma^2 n} + \lambda B^2} \quad \text{(Existing bound)}.$$

holds. The third inequality holds strictly when $\lambda > 0$.

For nonlinear case (i.e., $b_2 \neq 0$), we have achieved a tighter error bound. They proposed the following bound (23).

$$\|\Delta\|_{\Sigma_D(\theta^*)+\lambda I_d} \leq \alpha_1 \cdot \sqrt{\frac{d + \log(1/\delta)}{\alpha_2^2 n} + \left(\lambda + \frac{b_2}{\alpha_2} + b_1 b_2 B\right) B^2}, \tag{23}$$

where $\alpha_1$ and $\alpha_2$ are some universal constants. If $b_2 \neq 0$, as $n$ increases, the right side of equation (23) will not converge to 0, but to a constant $\alpha_1\sqrt{(\lambda + \frac{b_2}{\alpha_2} + b_1 b_2 B)B^2}$, even if $\lambda = 0$. However, our result at

Theorem 1 is tight. As the sample size $n$ approaches infinity, the right side of equation (14) tends to 0, since $a_0 \to 0$. Thus, comparing (14) and (23), there exists a value $\bar{N}'$, when $n > \bar{N}'$,

$$
\begin{aligned}
\|\Delta\|_{\Sigma_{\mathcal{D}}(\theta^*) + \lambda I_d} &\leq \frac{a_1 - \sqrt{a_1^2 - 4 * a_0 * a_2}}{2a_2} \\
&= \frac{a_1 - \sqrt{a_1^2 - 4 * \sqrt{\frac{d + 2\sqrt{d \log \frac{2}{\delta}} + 2 \log \frac{2}{\delta}}{n}} * a_2}}{2a_2} \ \text{(Our bound)} \\
&< \alpha_1 \cdot \sqrt{\frac{d + \log(1/\delta)}{\alpha_2^2 n} + \left( \lambda + \frac{b_2}{\alpha_2} + b_1 b_2 B \right) B^2} \ \text{(Existing bound)}
\end{aligned}
$$

holds. The third inequality will strictly hold, if $b_2 \neq 0$. Their bound under nonlinear conditions fails to answer whether parameter estimation errors would asymptotically approach zero with an infinite dataset, even under ideal conditions. In comparison, we propose a tighter bound that successfully fulfills this expectation.

The error bound proposed in Zhan et al. (2023a) involves the complexity of the reward function class, and the concentrability coefficient $C_{\text{coverage}}^2$, which reflects the quality of the data and is expressed as $C_{\text{coverage}}(\mathcal{R}, \pi^*, \mu_0, \mu_1) := \max \left\{ 0, \sup_{r \in \mathcal{R}} \frac{\mathbb{E}_{\tau_1 \sim \pi^*, \tau_2 \sim \mu_1}[r^*(\tau_1) - r^*(\tau_2) - r(\tau_1) + r(\tau_2)]}{\sqrt{\mathbb{E}_{\tau_1 \sim \mu_0, \tau_2 \sim \mu_1}[|r^*(\tau_1) - r^*(\tau_2) - r(\tau_1) + r(\tau_2)|^2]}} \right\}$, where $\mu_0$ and $\mu_1$ are two distributions of data sources. Compared to them, we do not require comprehensive coverage assumptions for the data collection policy, nor do we impose restrictions on the complexity of the reward function class. Although $C_{\text{coverage}}$ also describes the data quality, it involves the supremum operator, the expectation operator and the unattainable optimal parameters. These factors make $C_{\text{coverage}}$ difficult to compute and ineffective in guiding online data collection. Under ideal conditions, the proposed bound is computable and offers a theoretical approach for identifying an infinite dataset to achieve estimation errors approaching zero.

In the error bound proposed in Theorem 1, the most crucial parameter is $\underline{\alpha}$, which is the minimum eigenvalue of $\Sigma_D(\theta^*)$. If $\underline{\alpha} = 0$, then, the given bound is empty.

The case where $\underline{\alpha} = 0$ requires separate treatment. In this scenario, $\underline{\alpha} = 0$ allows $\Delta$ to become arbitrarily large in a certain subspace, which causes $\|\Delta\|_{\Sigma_D(\theta^*)}$ to remain bounded while $\|\Delta\|_{\Sigma_D(\theta^*) + \lambda I}$ grows without bound. Consequently, any discussion of error bounds becomes meaningless, including (22). It is reasonable and consistent with observations in existing studies. We begin by discussing the importance of $\underline{\alpha}$ from the perspective of the estimation algorithm. Zhu et al. (2023) explored the estimation problem within the family of linear structure reward functions, where $r^*(\tau) = \phi(\tau)^{\mathrm{T}} \theta^*$, and $\nabla \eta_{\theta^*}^i = \phi(\tau_1) - \phi(\tau_2)$. They used a pessimistic learning strategy to make learning more robust. However, we use an intuitive example in Appendix 3 to illustrate that even under the linear structure assumption, the pessimistic learning strategy may not be able to obtain any effective performance guarantee.

Theorem 1 is related to $\theta^*$. When the characteristics of the offline dataset satisfy the corresponding conditions, the estimation error is bounded and tends to 0 as sample size $n$ tends to infinity. When the offline dataset is imperfect, although Theorem 1 provides a tighter bound than existing studies, since $\theta^*$ is inaccessible, it fails to theoretically offer a practical and efficient online data collection strategy.

To address the problem of imperfections in offline datasets, we propose Theorem 2. This theorem clarifies the relationship between the learned parameters and the error bounds, providing critical insights into both the scope of parameter learning and the objectives of online data collection. In the next section, we will propose two RLHF algorithms induced by Theorem 2.

# 6 Application of the Tighter Estimation Error Bound

The previous analysis gives tighter bounds for general functions under the standard RLHF. Inspired by the theoretical results, we add constraints to the standard RLHF framework to highlight the importance of $\underline{\alpha}$. To further optimize $\underline{\alpha}$, we propose a novel online RLHF approach that iteratively collects data and retrains parameters, supported by Theorem 2.

### 6.1 Offline Iterative RLHF

Online data collection can provide much better coverage than the offline dataset. While the study (Xiong et al., 2024) confirms this result, its scope is restricted to linear function classes. Theorem 2 links the learned parameters, sample properties, and estimation error bounds. This observation motivates us to employ constrained optimization on the parameters to ensure that the learned parameters remain within the confidence region of the pre-collected dataset.

We modify the optimization problem (11)-(12) as follows.

$$\arg\min_{\theta} \quad L(\theta), \tag{24}$$

$$\text{s.t.} \quad \frac{1}{n}\sum_{i=1}^{n}\nabla\eta_{\theta}^{i}(\nabla\eta_{\theta}^{i})^{\mathrm{T}} \geq \alpha_{\text{target}}I_d, \tag{25}$$

$$\theta \in \Theta, \tag{26}$$

where $\alpha_{\text{target}}$ is a predefined constant.

This implies that in solving the optimization problem, we limit the scope of acceptable solutions to represent the confidence region of the offline data. This constraint is directly derived from the error bounds. In Appendix A.1, we provide a toy example to demonstrate how this constraint effectively restricts the parameter solutions to a feasible space. The offline RLHF algorithm under parameter constraints is summarized in Algorithm 2.

---
**Algorithm 2** Offline RLHF

---
    **Input**: offline dataset $\mathcal{D}$.
    1. **MLE**: compute $r_{\hat{\theta}}$ by solving (24)-(26).
    2. **Standard RL**: find $\hat{\pi} = \arg\max J_{r_{\hat{\theta}}}(\pi)$.

---

It may be difficult for the offline dataset to satisfy the constraints. Given this limitation, we propose a novel online RLHF that iteratively performs data collection and parameter optimization.

### 6.2 Online Iterative RLHF

We aim for the collected data to cover the feature space represented by $\theta^*$, as shown in Theorem 1. However, we actually only have the learned parameters $\hat{\theta}$. Inspired by Theorem 2, we propose a bi-level optimization approach for the original constrained optimization problem (24-25). Overall, we use the parameters obtained from the (t-1)-th iteration to calculate the characteristics that the data collected in the t-th iteration should satisfy. For the $t$-th iteration, with $\hat{\theta}^t$ fixed, we aim to collect data to increase the confidence of $\hat{\theta}^t$, i.e., finding $\{\tau_1^{j,t}, \tau_2^{j,t}\}_{j=1}^m$ that satisfy

$$\arg\max_{\{\tau_1^{j,t}, \tau_2^{j,t}\}_{j=1}^m} \lambda_{\min}\Big(\Sigma_A^t(\hat{\theta}^t) + \Sigma_D^{t-1}(\hat{\theta}^t)\Big), \tag{27}$$

where $m$ is the sample size per iteration,

$$\Sigma_A^t(\hat{\theta}^t) = \frac{1}{m}\sum_{j=1}^m \nabla_{\hat{\theta}^t}(r(\tau_1^{j,t}) - r(\tau_2^{j,t}))(\nabla_{\hat{\theta}^t}(r(\tau_1^{j,t}) - r(\tau_2^{j,t}))^{\mathrm{T}}$$

is generated by new data, and $\Sigma_D^{t-1}(\hat{\theta}^t) = \frac{1}{n}\sum_{i=1}^n \nabla\eta_{\hat{\theta}^t}^i(\nabla\eta_{\hat{\theta}^t}^i)^{\mathrm{T}} + \sum_{o=1}^{t-1}\Sigma_A^o(\hat{\theta}^t)$ is generated before the $t-1$-th additional data collection.

Then, retrain $\theta^{t+1}$ according to (28)-(30) to find a better solution that satisfies the constraints.

$$\arg\min_{\theta} \quad L(\theta, \mathcal{D}^t), \tag{28}$$

$$\text{s.t.} \quad \Sigma_D^t(\theta) \geq \alpha_{\text{target}} I_d, \tag{29}$$

$$\theta \in \Theta, \tag{30}$$

where $\mathcal{D}^t$ is the dataset used for training at the $t+1$-th iteration. The corresponding online RLHF is summarized in Algorithm 3.

**Remark 1.** *The proposed online algorithm's bottlenecks stem from bi-level optimization and minimum eigenvalue calculation. It provides guidance for data collection and does not necessarily require multiple iterations. In the simple experiment 6.3, we show the results obtained from a single iteration, which already demonstrate superior performance compared to random sampling. For computing the minimum eigenvalue, if we select $m$ trajectories from a candidate set of size $M$, the number of possible choices is $\binom{M}{m}$, which makes exhaustive evaluation computationally prohibitive. A reasonable approximation is to adopt a greedy algorithm that selects at each step the trajectory that yields the largest increase in the minimum eigenvalue. For high-dimensional problems, computing the minimum eigenvalue itself is challenging, and iterative methods are typically adopted to obtain approximate solutions, such as the LOBPCG algorithm.*

**Remark 2.** *The parameter $\alpha_{target}$ in (29) is determined by both the dataset and the learned parameters. During data collection, the value of $\underline{\alpha}$ corresponding to the current dataset under the current parameter $\hat{\theta}^t$ can be computed, and $\alpha_{target}$ is then empirically set as $\alpha_{target} = \frac{\underline{\alpha}(\hat{\theta}^t)}{2}$ for the $(t+1)$-th iteration.*

---

**Algorithm 3** Online Iterative RLHF

---

**Input**: offline dataset $\mathcal{D}$, iteration count $T$, sample size $m$ per iteration.
1. $\mathcal{D}^0 = \mathcal{D}$.
2. compute $r_{\hat{\theta}^1}$ by solving (24)-(26).
3. **for** iteration $t = 1, ..., T-1$ **do**
    collect data $\{\tau_1^{j,t}, \tau_2^{j,t}\}_{j=1}^m$ according to (27).
    $\mathcal{D}^t = \mathcal{D}^{t-1} \cup \{\tau_1^{j,t}, \tau_2^{j,t}\}_{j=1}^m$.
    compute $r_{\hat{\theta}^{t+1}}$ by solving (28)-(30).
5: **end for**
6. **Standard RL**: find $\hat{\pi} = \arg\max J_{r_{\hat{\theta}^T}}(\pi)$.

---

## 6.3 Experiment

This section evaluates the proposed algorithm on a small-scale reinforcement learning task and uses HalfCheetah-v4 from MuJoCo as the evaluation environment. The environment's default reward function is denoted as $r^*$. A preference dataset is then constructed by sampling a set of state pairs and assigning preferences according to $r^*$. The reward function maps the current state, next state, and action to a proxy reward value, comprising a total of 84 learnable parameters due to the omission of the bias term in the output layer.

We constructed $10,000$ preference samples based on the $r^*$. Then, for reward learning, 100 samples were initially selected to train the model $r_{\hat{\theta}^1}$, whose performance is evaluated using accuracy, Pearson, and Spearman correlation coefficients. After the initial training, we further collected 50 trajectories through random sampling and Algorithm 3 to obtain the proxy reward models $r_{\hat{\theta}^2}$ and $r_{\hat{\theta}^2}^{\text{Alg3}}$, respectively. A comparative analysis of these models is provided in Table 2.

Table 2: Evaluation metrics on the overall preference dataset.

| Proxy reward model | Accuracy | Pearson coefficient | Spearman coefficient |
|:---:|:---:|:---:|:---:|
| $r_{\hat{\theta}^1}$ | 73.39% | 0.5666 | 0.6616 |
| $r_{\hat{\theta}^2}$ | 77.43% | 0.6789 | 0.7348 |
| $r_{\hat{\theta}^2}^{\text{Alg3}}$ | **96.43%** | **0.9906** | **0.9929** |

We adopt TD3 as the downstream reinforcement learning algorithm. Fig. 1 shows the performance of four different reward functions ($r^*$, $r_{\hat{\theta}^1}$, $r_{\hat{\theta}^2}$ and $r_{\hat{\theta}^2}^{\text{Alg3}}$) in downstream DRL tasks, where the y-axis represents the true and unnormalized reward. It can be seen that the proposed online collection algorithm improves data collection efficiency and achieves performance on downstream tasks that is closer to training with ground-truth reward compared to random sampling.

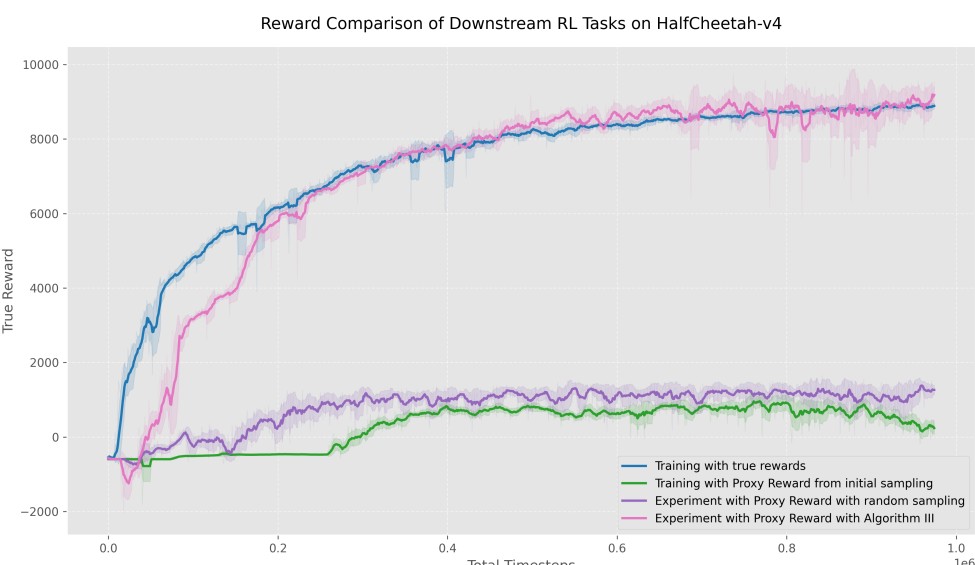

Figure 1: Comparison of Downstream RL Tasks on HalfCheetah-v4

# 7 Conclusion

We have provided a theoretical convergence analysis of RLHF under general function approximation. The main results involve two insights: (i) when the preference dataset satisfies some conditions, the parameter estimation error converges to 0 without any constraints on the complexity of the reward function class (Theorem 1); and (ii) collecting data that enriches the gradient information of the reward function is beneficial to parameter learning (Algorithm 3).

Although we have made progress in understanding the convergence of RLHF, several issues remain unresolved.

1). Although it is not difficult to find the gradient of the reward function, finding the minimum eigenvalue of a matrix is computationally complex, and how to find an effective approximation remains an open issue.

2). This paper studies the PAC problem. If the optimal parameters are not in the solution space, whether a meaningful error bound can still be guaranteed remains an open issue.

**Acknowledgments**

This research is supported by National Natural Science Foundation (NNSF) of China under Grant 62373100 and 62233003, National Science and Technology Major Project under Grant 2021ZD0112702, and SEU Innovation Capability Enhancement Plan for Doctoral Students under Grant CXJH_SEU 24131. This research is supported by the Ministry of Education, Singapore, under its MOE AcRF Tier 2 Award MOE-T2EP20223-0003. Any opinions, findings and conclusions or recommendations expressed in this material are those of the author(s) and do not reflect the views of the Ministry of Education, Singapore.

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

# A  Toy example

## A.1  Toy Example 1

This example is used to verify the proposed offline RLHF.

**Example 1.** *Consider a simple nonlinear reward function learning problem with only 4 trajectories, denoted as*

$$\tau_1 = \begin{bmatrix} 0 \\ 0 \end{bmatrix}, \tau_2 = \begin{bmatrix} 0 \\ 1 \end{bmatrix}, \tau_3 = \begin{bmatrix} 1 \\ 0 \end{bmatrix}, \ and \ \tau_4 = \begin{bmatrix} 1 \\ 1 \end{bmatrix}.$$

*Let the ground truth reward be parameterized by $\theta^* = \begin{bmatrix} 1 \\ -1.1 \end{bmatrix}$, and the nonlinear function defined as*

$$r^*(\tau) = \phi(\tau)^{\mathrm{T}}\theta^* + \theta^*[1] * \theta^*[2] * \tau[1] * \tau[2] + (\theta^*[1])^3\tau[1] + (\theta^*[2])^3\tau[2],$$

*where $\theta^*[1]$ and $\theta^*[2]$ represent the first and second elements of $\theta^*$, respectively, and $\tau[1]$ and $\tau[2]$ denote the first and second elements of $\tau$, respectively. Then,*

$$\triangledown r_\theta(\tau) = \begin{bmatrix} \theta^*[2] * \tau[1] * \tau[2] + \tau[1] + 3(\theta^*[1])^2\tau[1] \\ \theta^*[1] * \tau[1] * \tau[2] + \tau[2] + 3(\theta^*[2])^2\tau[2] \end{bmatrix}.$$

*Although constrained optimization problems are unlikely to yield solutions that make the objective function smaller than unconstrained optimization problems, when there are multiple solutions, constraints can eliminate unreliable solutions suggested by the data. Fig. 2 shows this point. Assuming uniform sampling, the MLE solution may be inaccurate when the sample size is small. In this case, the solution that minimizes the loss function (5) becomes invalid, resulting in erroneous estimations. Fortunately, the constraint helps mitigate the impact of this error.*

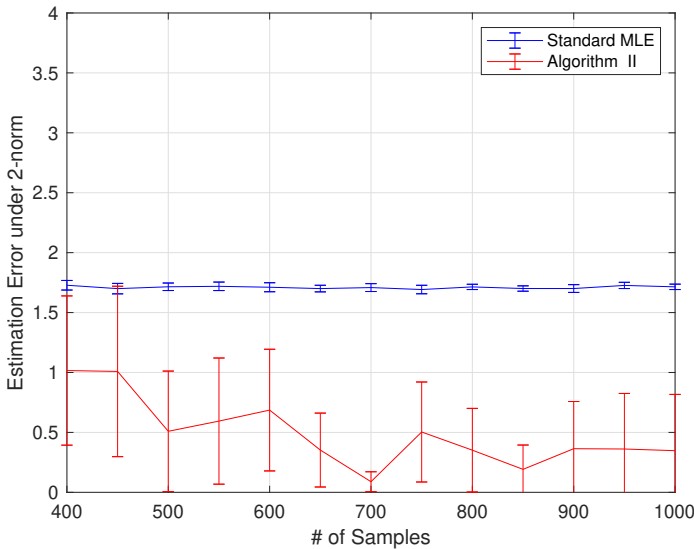

Figure 2: Comparison of estimation errors versus # of samples.

## A.2 Toy Example 2

This example is a toy verification of the proposed online RLHF. Consider a simple nonlinear reward function learning problem with only 4 trajectories, denoted as $\tau_1 = \begin{bmatrix} 0 \\ 0 \end{bmatrix}, \tau_2 = \begin{bmatrix} 0 \\ 1 \end{bmatrix}, \tau_3 = \begin{bmatrix} 1 \\ 0 \end{bmatrix}$, and $\tau_4 = \begin{bmatrix} 1 \\ 1 \end{bmatrix}$. Let the ground truth reward be parameterized by $\theta^* = \begin{bmatrix} 1 \\ -1.1 \end{bmatrix}$, and the nonlinear function is defined as $r^*(\tau) = \phi(\tau)^{\mathrm{T}}\theta^* + \theta^*[1] * \theta^*[2] * \tau[1] * \tau[2]$, where $\theta^*[1]$ and $\theta^*[2]$ represent the first and second elements of $\theta^*$, respectively, and $\tau[1]$ and $\tau[2]$ denote the first and second elements of $\tau$, respectively. Then, $\nabla r_\theta(\tau) = \begin{bmatrix} \theta^*[2] * \tau[1] * \tau[2] + \tau[1] \\ \theta^*[1] * \tau[1] * \tau[2] + \tau[2] \end{bmatrix}$. We evaluate the effectiveness of Algorithm 3 by comparing the estimation error of standard RLHF, which uses uniform sampling, with that of Algorithm 3 across varying sample sizes. We mainly focus on the case when the number of samples is small (less than 100). We set $T = 2$ in Algorithm 3, and each sample size is tested 10 times. When deploying Algorithm 3, the first $N/2$ samples are obtained through uniform sampling. Then, the parameters are obtained by solving the constrained optimization problems (28)-(30) and further data collection is based on the learned parameters and (27). The comparative experimental result is presented in Fig. 3.

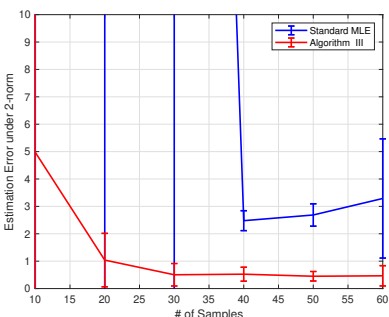

Figure 3: Comparison of estimation errors versus # of samples.

Compared to uniform sampling, Algorithm 3 achieves better estimation results when the sample size is small. This is due to the fact that the latter $N/2$ samples provide the necessary information for more accurate parameter estimation with respect to the learned parameters. These experimental results suggest that the proposed tighter bound offers a more effective strategy for online data collection.

### A.3 Toy Example 3

This example aims to present the curve of $\|\frac{1}{n}\sum_{i=1}^{n}\left(p_i * \frac{1}{1+e^{\eta_\theta^i}} - (1-p_i)\frac{e^{\eta_\theta^i}}{1+e^{\eta_\theta^i}}\right)\nabla^2\eta_{\hat\theta}^i\|_2$ as a function of $\sqrt{n}$, illustrating how we can compress the existing bounds.

**Example 2.** *We use the same configuration as the Example 1 and eliminated randomness by averaging the results from* $100$ *experiments. The relationship between* $\|\frac{1}{n}\sum_{i=1}^{n}\left(p_i * \frac{1}{1+e^{\eta_\theta^i}} - (1-p_i)\frac{e^{\eta_\theta^i}}{1+e^{\eta_\theta^i}}\right)\nabla^2\eta_{\hat\theta}^i\|_2$ *and* $\sqrt{n}$ *as shown in Fig. 4. is shown in Fig. 4. From Fig. 4, we realized that although* $\nabla^2\eta_{\hat\theta}^i\|_2$ *is related to* $\hat\theta$

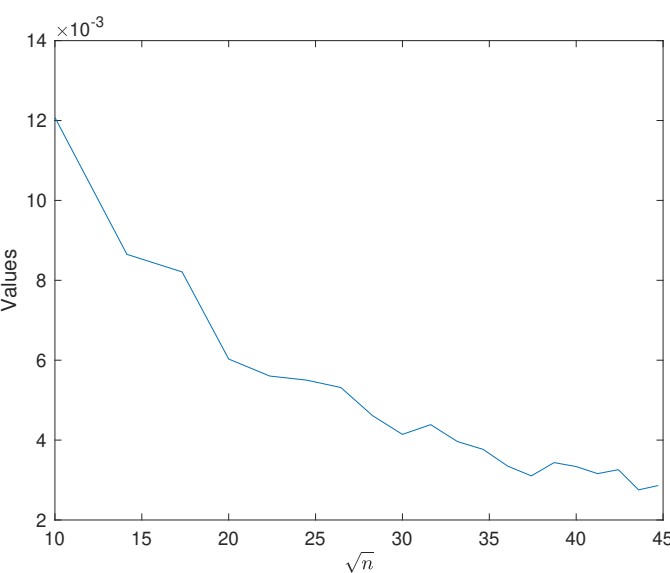

Figure 4: The relationship between a key term in the bound and $\sqrt{n}$.

*and therefore to* $p_i * \frac{1}{1+e^{\eta_\theta^i}} - (1-p_i)\frac{e^{\eta_\theta^i}}{1+e^{\eta_\theta^i}}$. $\|\frac{1}{n}\sum_{i=1}^{n}\left(p_i * \frac{1}{1+e^{\eta_\theta^i}} - (1-p_i)\frac{e^{\eta_\theta^i}}{1+e^{\eta_\theta^i}}\right)\nabla^2\eta_{\hat\theta}^i\|_2$ *may still approach zero. In fact, Fig. 4 shows that it approaches* $0$ *at a rate of approximately* $\sqrt{n}$*. This intuition encourages us to compress the existing bounds and derive the tighter theoretical results.*

### A.4 Toy Example 4

**Example 3.** *In this example, we will show that even a pessimistic learning strategy cannot achieve any meaningful results if* $\underline{\alpha} = 0$*.*

*Consider* $4$ *trajectories encoded by* $\phi(\tau_1) = \begin{bmatrix} 1 \\ 1 \\ 0 \end{bmatrix}, \phi(\tau_2) = \begin{bmatrix} 0 \\ 1 \\ 0 \end{bmatrix}, \phi(\tau_3) = \begin{bmatrix} 0 \\ 0 \\ 0 \end{bmatrix}$*, and* $\phi(\tau_4) = \begin{bmatrix} 0 \\ 0 \\ 1 \end{bmatrix}$*. The pre-collected dataset contains* $\frac{n}{2}$ *comparison results of* $\tau_1$ *and* $\tau_2$ *and* $\frac{n}{2}$ *comparison results of* $\tau_2$ *and* $\tau_3$*. If the reward function class follows a linear structure, i.e.,* $r^*(\tau_i) = \phi(\tau_i)^{\mathrm{T}}\theta^*$*, then,* $\Sigma_D(\theta^*) = \begin{bmatrix} 0.5 & 0 & 0 \\ 0 & 0.5 & 0 \\ 0 & 0 & 0 \end{bmatrix}$ *has*

the minimum eigenvalue $0$. The parameters to be learned are written as $\theta = \begin{bmatrix} \theta_1 \\ \theta_2 \\ \theta_3 \end{bmatrix}$. Divide the dataset into two parts $\{\tau_1^i, \tau_2^i, p_i\}_{i=1}^{\frac{n}{2}}$, and $\{\tau_2^i, \tau_3^i, p_i'\}_{i=1}^{\frac{n}{2}}$ then the loss (5) can be reorganized as

$$L(\theta) = -\frac{1}{n}\left[ \sum_{i=1}^{\frac{n}{2}} p_i \log \frac{1}{1+e^{-\theta_1}} + (1-p_i) \log \frac{1}{1+e^{\theta_1}} \right.$$
$$\left. + \sum_{i=1}^{\frac{n}{2}} p_i' \log \frac{1}{1+e^{-\theta_2}} + (1-p_i') \log \frac{1}{1+e^{\theta_2}} \right].$$

It is easy to verify that for a MLE solution $\hat{\theta} = \begin{bmatrix} \hat{\theta}_1 \\ \hat{\theta}_2 \\ \hat{\theta}_3 \end{bmatrix}$, parameters $\begin{bmatrix} \hat{\theta}_1 \\ \hat{\theta}_2 \\ \bar{\theta} \end{bmatrix}$ with any $\bar{\theta}$ is also a MLE solution.

Note that the Pessimistic MLE of Zhu et al. (2023) try to find the optimal strategy as follows.

$$\arg \max_{\pi} \left[ \arg \min_{\theta \in \Theta(\hat{\theta}, \lambda)} J_{r_\theta}(\pi) \right]. \tag{31}$$

Let the ground truth reward be $\theta^* = [0.1, -0.1, 1]$, the optimal strategy should generates $\tau_4$ with probability $1$. However, the optimal solution from (31) produces $\tau_4$ with probability $0$, since

$$\begin{bmatrix} \hat{\theta}_1 \\ \hat{\theta}_2 \\ \sqrt{B - \hat{\theta}_1^2 - \hat{\theta}_2^2} \end{bmatrix} \in \Theta(\hat{\theta}, \lambda).$$

This suggests that the optimal policy learned based on a pessimistic optimization objective (31) can still exhibit an arbitrarily large performance gap compared to the true optimal policy when $\underline{\alpha} = 0$.

## B  Auxiliary Lemmas

**Lemma 1.** *(Objective decomposition). Let $J(\pi)$ be the objective function defined in (1), and for any reward function $\hat{r}$, the gap, as expressed by 2, between the induced optimal policy $\hat{\pi} = \arg \max_\pi J_{\hat{r}}(\pi)$ and the optimal performance can be reformulated as follows.*

$$G(\pi^*, \hat{\pi}) = \mathbb{E}_{s \sim \rho} \mathbb{E}_{\tau \sim \pi^*}[r^*(\tau) - \hat{r}(\tau)] + \mathbb{E}_{s \sim \rho} \mathbb{E}_{\tau \sim \hat{\pi}}[\hat{r}(\tau) - r^*(\tau)] \tag{32}$$

*Proof.* This proof mainly refers to Lemma A.1 of Song et al. (2024). Note that $\hat{\pi} = \arg \max_\pi J_{\hat{r}}(\pi)$, thus

$$J_{\hat{r}}(\hat{\pi}) \geq J_{\hat{r}}(\pi^*).$$

Then, we have

$$\begin{aligned} G(\pi^*, \hat{\pi}) &= J_{r^*}(\pi^*) - J_{r^*}(\hat{\pi}) \\ &= \mathbb{E}_{s \sim \rho} \mathbb{R}_{\tau \sim \pi^*}[r^*(\tau)] - \mathbb{E}_{s \sim \rho} \mathbb{E}_{\tau \sim \hat{\pi}}[r^*(\tau)] \\ &= \mathbb{E}_{s \sim \rho} \mathbb{E}_{\tau \sim \pi^*}[r^*(\tau)] - \mathbb{E}_{s \sim \rho} \mathbb{E}_{\tau \sim \hat{\pi}}[r^*(\tau)] + \mathbb{E}_{s \sim \rho} \mathbb{E}_{\tau \sim \hat{\pi}}[\hat{r}(\tau)] - \mathbb{E}_{s \sim \rho} \mathbb{E}_{\tau \sim \hat{\pi}}[\hat{r}(\tau)] \\ &\leq \mathbb{E}_{s \sim \rho} \mathbb{E}_{\tau \sim \pi^*}[r^*(\tau)] - \mathbb{E}_{s \sim \rho} \mathbb{E}_{\tau \sim \pi^*}[\hat{r}(\tau)] + \mathbb{E}_{s \sim \rho} \mathbb{E}_{\tau \sim \hat{\pi}}[\hat{r}(\tau)] - \mathbb{E}_{s \sim \rho} \mathbb{E}_{\tau \sim \hat{\pi}}[r^*(\tau)] \\ &= \mathbb{E}_{s \sim \rho} \mathbb{E}_{\tau \sim \pi^*}[r^*(\tau) - \hat{r}(\tau)] + \mathbb{E}_{s \sim \rho} \mathbb{E}_{\tau \sim \hat{\pi}}[\hat{r}(\tau) - r^*(\tau)]. \end{aligned} \tag{33}$$

This completes the proof. $\qquad\square$

**Lemma 2.** *(Hsu et al., 2012) Let $M \in \mathbb{R}^{n \times n}$ be a matrix. Suppose that $y = [y_1, ..., y_n]^{\mathrm{T}}$ is a random vector such that,*

$$\mathbb{E}\left[e^{\alpha^{\mathrm{T}}y}\right] \leq e^{\|\alpha\|_2^2/2} \tag{34}$$

*holds for all $\alpha \in \mathbb{R}^n$. For all $t > 0$, the following probability inequality holds.*

$$\mathbb{P}\left[y^{\mathrm{T}}My > \mathrm{tr}(M) + 2\sqrt{\mathrm{tr}\left(M^2\right)t} + 2\|M\|t\right] \leq e^{-t} \tag{35}$$

**Lemma 3.** *For a matrix $A \in \mathbb{R}^{d \times d}$, assuming its elements are bounded by $|a_{ij}| \leq b$, then, the spectral norm of $A$ is bounded as follows:*

$$\|A\|_2 \leq db. \tag{36}$$

*Proof.* According to the properties of matrix norms, the spectral norm of a matrix $A$ is bounded above by its Frobenius norm $\|A\|_F$:

$$\|A\|_2 \leq \|A\|_F. \tag{37}$$

The Frobenius norm is defined as the square root of the sum of the absolute squares of its elements:

$$\|A\|_F = \sqrt{\sum_{i=1}^{d}\sum_{j=1}^{d}|a_{ij}|^2} \tag{38}$$

Given the condition that each element is bounded by $|a_{ij}| \leq b$, we can derive the following inequality:

$$\sum_{i=1}^{d}\sum_{j=1}^{d}|a_{ij}|^2 \leq \sum_{i=1}^{d}\sum_{j=1}^{d}b^2 = d^2b^2 \tag{39}$$

Taking the square root of both sides yields the upper bound for the Frobenius norm:

$$\|A\|_F \leq \sqrt{d^2b^2} = db. \tag{40}$$

Consequently, applying the relationship between the spectral norm and the Frobenius norm, we obtain:

$$\|A\|_2 \leq \|A\|_F \leq db \tag{41}$$

This completes the proof. $\square$

## C   Proof of the Theorem

### C.1   Proof of the Theorem 1

**Theorem 1** (Restatement). *Under the Assumption 2-5, if $\underline{\alpha} > 0$, for any $\lambda \geq 0$, with probability $1 - \delta$, (1) When $c_1 = 0$, we have*

$$\|\Delta\|_{\Sigma_D(\theta^*)+\lambda I_d} \leq \frac{a_0}{a_1}. \tag{42}$$

*(2) When $c_1 > 0$, $\exists n^*$, after $n > n^*$, we have*

$$\|\Delta\|_{\Sigma_D(\theta^*)+\lambda I_d} \leq \frac{a_1 - \sqrt{a_1^2 - 4*a_0*a_2}}{2a_2}, \tag{43}$$

*if* $B < \frac{a_1 + \sqrt{a_1^2 - 4 * a_0 * a_2}}{4 a_2 \sqrt{4 b_1^2 + \lambda}}$, *where*

$$a_2 = \underline{\beta}^3 (8 \zeta b_1 b_2 + b_1 c_1 d + 2 c_2 d)$$

$$a_1 = \zeta - \underline{\beta}^2 (\zeta \lambda + d c_1 \sqrt{\frac{8 \log(2 d^2 / \delta)}{n}}),$$

$$a_0 = \sqrt{\frac{d + 2 \sqrt{d \log \frac{2}{\delta}} + 2 \log \frac{2}{\delta}}{n}},$$

$\underline{\beta} = \frac{1}{\sqrt{\lambda + \underline{\alpha}}}$, *and* $\underline{\alpha}$ *is the minimum eigenvalue of* $\Sigma_D(\theta^*)$, *with* $\Sigma_D(\theta^*) = \frac{1}{n} \sum_{i=1}^n \nabla \eta_{\theta*}^i (\nabla \eta_{\theta*}^i)^{\mathrm{T}}$.

*Proof.* First, we get the expression of $\nabla l(\theta) \in \mathbb{R}^d$ as follows

$$
\begin{aligned}
\nabla l(\theta) &= -\frac{1}{n} \left[ \sum_{i=1}^n \left( p_i * \frac{e^{-\eta_\theta^i} * \nabla \eta_\theta^i}{1 + e^{-\eta_\theta^i}} - (1 - p_i) \frac{e^{\eta_\theta^i} * \nabla \eta_\theta^i}{1 + e^{\eta_\theta^i}} \right) \right] \\
&= -\frac{1}{n} \left[ \sum_{i=1}^n \left( p_i * \frac{e^{-\eta_\theta^i}}{1 + e^{-\eta_\theta^i}} - (1 - p_i) \frac{e^{\eta_\theta^i}}{1 + e^{\eta_\theta^i}} \right) \nabla \eta_\theta^i \right] \\
&= -\frac{1}{n} \left[ \sum_{i=1}^n \left( p_i * \frac{1}{1 + e^{\eta_\theta^i}} - (1 - p_i) \frac{e^{\eta_\theta^i}}{1 + e^{\eta_\theta^i}} \right) \nabla \eta_\theta^i \right].
\end{aligned}
\tag{44}
$$

The next step is to compute the Hessian matrix, $\nabla^2 l(\theta) \in \mathbb{R}^{d \times d}$:

$$
\begin{aligned}
\nabla^2 l(\theta) &= -\frac{1}{n} \left[ \sum_{i=1}^n \left( p_i * \frac{1}{1 + e^{\eta_\theta^i}} - (1 - p_i) \frac{e^{\eta_\theta^i}}{1 + e^{\eta_\theta^i}} \right) \nabla^2 \eta_\theta^i \right] \\
&\quad - \frac{1}{n} \left[ \sum_{i=1}^n \left( p_i * \frac{-e^{\eta_\theta^i}}{(1 + e^{\eta_\theta^i})^2} - (1 - p_i) \frac{e^{\eta_\theta^i}}{1 + e^{\eta_\theta^i}} + (1 - p_i) \frac{(e^{\eta_\theta^i})^2}{(1 + e^{\eta_\theta^i})^2} \right) \nabla \eta_\theta^i (\nabla \eta_\theta^i)^{\mathrm{T}} \right] \\
&= g_1(\theta) + g_2(\theta).
\end{aligned}
\tag{45}
$$

where

$$g_1(\theta) = -\frac{1}{n} \sum_{i=1}^n \left( p_i * \frac{1}{1 + e^{\eta_\theta^i}} - (1 - p_i) \frac{e^{\eta_\theta^i}}{1 + e^{\eta_\theta^i}} \right) \nabla^2 \eta_\theta^i. \tag{46}$$

and

$$g_2(\theta) = \frac{1}{n} \sum_{i=1}^n \left( \frac{e^{\eta_\theta^i}}{(1 + e^{\eta_\theta^i})^2} \nabla \eta_\theta^i (\nabla \eta_\theta^i)^{\mathrm{T}} \right). \tag{47}$$

Then, we need to bound these two items respectively.

For convenience, noting that in the subsequent analysis we use the variable $\eta_\theta^i = r_\theta(\tau_1^i) - r_\theta(\tau_2^i)$, we first convert the constraints on the reward function in Assumption 3 into constraints on $\eta_\theta^i$. For any $\theta \in \Theta$, $\tau_1, \tau_2 \in \mathcal{T}$, the following inequalities are hold according to the triangle inequality of norms (Horn & Johnson,

2012):

$$|\eta_\theta(\tau_1, \tau_2)| = |r_\theta(\tau_1) - r_\theta(\tau_2)| \le |r_\theta(\tau_1)| + |r_\theta(\tau_2)| \le 2b_0 \tag{48}$$

$$\|\nabla\eta_\theta(\tau_1, \tau_2)\|_2 = \|\nabla r_\theta(\tau_1) - \nabla r_\theta(\tau_2)\|_2 \le \|\nabla r_\theta(\tau_1)\|_2 + \|\nabla r_\theta(\tau_2)\|_2 \le 2b_1 \tag{49}$$

$$\|\nabla^2\eta_\theta(\tau_1, \tau_2)\|_2 = \|\nabla^2 r_\theta(\tau_1) - \nabla^2 r_\theta(\tau_2)\|_2 \le \|\nabla^2 r_\theta(\tau_1)\|_2 + \|\nabla^2 r_\theta(\tau_2)\|_2 \le 2b_2 \tag{50}$$

$$|\frac{\partial^2\eta_\theta(\tau_1, \tau_2)}{\theta_i\theta_j}| = |\frac{\partial^2 r_\theta(\tau_1)}{\theta_i\theta_j} - \frac{\partial^2 r_\theta(\tau_2)}{\theta_i\theta_j}| \le 2c_1 \tag{51}$$

$$\|\nabla\frac{\partial^2\eta_\theta(\tau_1, \tau_2)}{\theta_i\theta_j}\|_2 = \|\nabla\frac{\partial^2 r_\theta(\tau_1)}{\theta_i\theta_j} - \nabla\frac{\partial^2 r_\theta(\tau_2)}{\theta_i\theta_j}\|_2 \le 2c_2, \tag{52}$$

Next, we will discuss the boundaries of $g_1(\theta)$ and $g_2(\theta)$, respectively.

**(1).** For $g_1(\theta)$, we discuss the bound of $g_1(\theta')$ in the range of $\theta' = \theta^* + \sigma\Delta, \sigma \in [0, 1]$. Recall that $\Delta = \hat\theta - \theta^*$. Intuitively, we leverage the continuity of $r_\theta \in \mathcal{R}$ with respect to $\theta \in \Theta$ to explore the property that $\theta'$ satisfies, where $\theta'$ lies between $\theta^*$ and $\hat\theta$.

Then,

$$g_1(\theta') = -\frac{1}{n}\sum_{i=1}^n \left(p_i * \frac{1}{1+e^{\eta_{\theta'}^i}} - (1-p_i)\frac{e^{\eta_{\theta'}^i}}{1+e^{\eta_{\theta'}^i}}\right)\nabla^2\eta_{\theta'}^i.$$

Next, we aim to bound matrix $g_1(\theta') \in \mathbb{R}^{d\times d}$ using Lemma 3, and therefore perform an element-wise analysis of $g_1(\theta')$ first.

In order to reduce the pressure of symbolic expression, we define the elements of the matrix $\nabla^2\eta_{\theta'}^i$ as $a'^i_{jk}$, where $i \le n$, and $1 \le j, k \le d$. The apostrophe indicates that this element is associated with $\theta'$. From (52), we get $-2c_1 \le a'^i_{jk} \le 2c_1$.

Then, the elements of the matrix $g_1(\theta')$ can be expressed as

$$g'_{jk} = -\frac{1}{n}\sum_{i=1}^n g'^i_{jk}, \tag{53}$$

where $g'^i_{jk} = \left(p_i * \frac{1}{1+e^{\eta_{\theta'}^i}} - (1-p_i)\frac{e^{\eta_{\theta'}^i}}{1+e^{\eta_{\theta'}^i}}\right)a'^i_{jk}$. Decompose $g'_{jk}$ into two parts as shown in (54): $g'_{jk} - g^*_{jk}$ and $g^*_{jk}$. The first term is associated with $\theta' - \theta^*$, while the other involves $\theta^*$ and is expected to be 0.

$$g'_{jk} = (g'_{jk} - g^*_{jk}) + g^*_{jk}, \tag{54}$$

where $g^*_{jk} = -\frac{1}{n}\sum_{i=1}^n g^{*i}_{jk}$, and

$$g^{*i}_{jk} = \left(p_i * \frac{1}{1+e^{\eta_{\theta^*}^i}} - (1-p_i)\frac{e^{\eta_{\theta^*}^i}}{1+e^{\eta_{\theta^*}^i}}\right)a^{*i}_{jk}. \tag{55}$$

Next, we need to find bounds for $(g'_{jk} - g^*_{jk})$ and $g^*_{jk}$ respectively.

We will begin by finding the bound for $g^*_{jk}$. After fixing the samples $\tau_1^i$ and $\tau_2^i$, $g^{*i}_{jk}$ is a random variable, with its randomness arising from $p_i$. Define an auxiliary random variable

$$y_i = p_i * \frac{1}{1+e^{\eta_{\theta^*}^i}} - (1-p_i)\frac{e^{\eta_{\theta^*}^i}}{1+e^{\eta_{\theta^*}^i}}. \tag{56}$$

Then, take the expectation on both sides of equation (56), we have

$$\mathbb{E}[y_i] = \mathbb{E}\left[p_i * \frac{1}{1+e^{\eta_{\theta*}^i}}\right] - \mathbb{E}\left[(1-p_i)\frac{e^{\eta_{\theta*}^i}}{1+e^{\eta_{\theta*}^i}}\right]. \tag{57}$$

Note that the randomness on the right side of the equation (57) comes from $p_i$, and $\frac{e^{\eta_{\theta*}^i}}{1+e^{\eta_{\theta*}^i}}$ and $\frac{1}{1+e^{\eta_{\theta*}^i}}$ are constants. We can further obtain

$$\mathbb{E}[y_i] = \mathbb{E}[p_i] * \frac{1}{1+e^{\eta_{\theta*}^i}} - (1-\mathbb{E}[p_i])\frac{e^{\eta_{\theta*}^i}}{1+e^{\eta_{\theta*}^i}} = 0. \tag{58}$$

Similarly, since $a^{*i}_{jk}$ is independent of $p_i$,

$$\mathbb{E}[g^{*i}_{jk}] = \mathbb{E}[y_i]a^{*i}_{jk} = 0 \tag{59}$$

Since $|y_i| <= 1$, $g^{*i}_{jk}$ is also a bounded random variable, with $-2c_1 \le g^{*i}_{jk} \le 2c_1$. Thus, according to Hoeffding's inequality, we have

$$\mathbb{P}(|g^*_{jk}| \ge t) = \mathbb{P}(\left|\frac{1}{n}\sum_{i=1}^n g^{*i}_{jk} - \mathbb{E}[\frac{1}{n}\sum_{i=1}^n g^{*i}_{jk}]\right| \ge t) \le 2e^{\left(-\frac{2n^2t^2}{\sum_{i=1}^n (4c_1)^2}\right)}. \tag{60}$$

Simplify (60) and get

$$\mathbb{P}(|g^*_{jk}| \ge t) \le 2e^{\frac{-nt^2}{8c_1{}^2}}. \tag{61}$$

Rewriting (61) gives

$$\mathbb{P}(|g^*_{jk}| \le t) \ge 1 - 2e^{\frac{-nt^2}{8c_1{}^2}}. \tag{62}$$

Now, let

$$\delta = 2e^{\frac{-nt^2}{8c_1{}^2}}, \tag{63}$$

which leads to

$$t = c_1\sqrt{\frac{8\log(2/\delta)}{n}}. \tag{64}$$

Thus, $|g^*_{jk}|$ can be bounded with the probability of $1-\delta$

$$|g^*_{jk}| \le c_1\sqrt{\frac{8\log(2/\delta)}{n}}, \tag{65}$$

for some special $j, k$.

Then, consider the bound of another part, $g'_{jk} - g^*_{jk}$. The idea behind proving its boundedness is to use the Lipschitz continuity condition to transform it into a bound related to the estimation error $\theta^* - \theta'$.

$$g'_{jk} - g^*_{jk} = -\frac{1}{n}\sum_{i=1}^n \left(p_i * \frac{1}{1+e^{\eta_{\theta'}^i}} - (1-p_i)\frac{e^{\eta_{\theta'}^i}}{1+e^{\eta_{\theta'}^i}}\right)a'^i_{jk} + \frac{1}{n}\sum_{i=1}^n \left(p_i * \frac{1}{1+e^{\eta_{\theta*}^i}} - (1-p_i)\frac{e^{\eta_{\theta*}^i}}{1+e^{\eta_{\theta*}^i}}\right)a^{*i}_{jk}, \tag{66}$$

Consider an auxiliary function

$$f_{jk}^i(\theta) = \left( p_i * \frac{1}{1 + e^{\eta_\theta^i}} - (1 - p_i) \frac{e^{\eta_\theta^i}}{1 + e^{\eta_\theta^i}} \right) \frac{\partial^2 \eta_\theta(\tau)}{\theta_j \theta_k} \tag{67}$$
$$= f_1^i(\theta) f_2^i(\theta).$$

Note that

$$\nabla f_1^i(\theta) = \frac{\partial f_1^i(\theta)}{\partial \theta} = p_i * \frac{-e^{\eta_\theta^i}}{(1 + e^{\eta_\theta^i})^2} * \nabla \eta_\theta(\tau_1^i, \tau_2^i) - (1 - p_i) \frac{e^{\eta_\theta^i}}{(1 + e^{\eta_\theta^i})^2} * \nabla \eta_\theta(\tau_1^i, \tau_2^i), \tag{68}$$

according to (50), we have

$$\|\nabla f_1^i(\theta)\|_2 \le 0.5 b_1, \tag{69}$$

since $|\frac{-e^x}{(1+e^x)^2}| \le 0.25$ and $|\frac{e^x}{(1+e^x)^2}| \le 0.25$ always holds for any $x$.

Thus,

$$\nabla f_{jk}^i(\theta) = \nabla f_1^i(\theta) f_2^i(\theta) + f_1^i(\theta) \nabla f_2^i(\theta). \tag{70}$$

Furthermore, we have

$$\|\nabla f_{jk}^i(\theta)\|_2 \le \|\nabla f_1^i(\theta) f_2^i(\theta)\|_2 + \|f_1^i(\theta) \nabla f_2^i(\theta)\|_2 \le b_1 c_1 + 2 c_2, \tag{71}$$

since $|\frac{1}{1+e^x}| \le 1$ and $|\frac{e^x}{1+e^x}| \le 1$ always holds for any $x > 0$.

We further know that $f_{jk}^i(\theta)$ is continuous with respect to $\theta$, since $f_1^i(\theta)$ and $f_2^i(\theta)$ is continuous according to the assumptions. According to the mean value theorem, we have

$$f^i(\theta') - f^i(\theta^*) = \nabla f^i(\bar{\theta})^{\mathrm{T}}(\theta' - \theta^*)$$

holds for some $\bar{\theta}$. Thus,

$$|f^i(\theta') - f^i(\theta^*)| = |\nabla f^i(\bar{\theta})^{\mathrm{T}}(\theta' - \theta^*)| \le \|\nabla f^i(\bar{\theta})\|_2 \|\theta' - \theta^*\|_2 \le (b_1 c_1 + 2 c_2) \|\theta' - \theta^*\|_2$$

holds for any $1 \le i \le n$, and $1 \le j, k \le d$. Combined with the above analysis, we have

$$|g_{jk}' - g_{jk}^*| = \left| \frac{1}{n} \sum_{i=1}^n \left( f_{jk}^i(\theta^*) - f_{jk}^i(\theta') \right) \right|$$
$$\le \frac{1}{n} \sum_{i=1}^n |f_{jk}^i(\theta^*) - f_{jk}^i(\theta')|$$
$$\le (b_1 c_1 + 2 c_2) \|\theta' - \theta^*\|_2$$

According to (54) and (65),

$$|g_{jk}'| \le (b_1 c_1 + 2 c_2) \|\theta' - \theta^*\|_2 + c_1 \sqrt{\frac{8 \log(2/\delta)}{n}} \tag{72}$$

holds with the probability of $1 - \delta$, for special $j, k$. Using union bound, with probability $1 - \delta$, we have

$$|g_{jk}'| \le (b_1 c_1 + 2 c_2) \|\theta' - \theta^*\|_2 + c_1 \sqrt{\frac{8 \log(2 d^2/\delta)}{n}} \tag{73}$$

for any $j, k$.

According to (73) and Lemma 3, We get the following results about the spectral norm of matrix $g_1(\theta')$. for any $\theta' = \theta^* + \sigma\Delta, \sigma \in [0,1]$, with the probability of $1 - \delta$, we have

$$g_1(\theta') \leq \left( d(b_1 c_1 + 2c_2)\|\theta' - \theta^*\|_2 + dc_1 \sqrt{\frac{8\log(2d^2/\delta)}{n}} \right) I_d. \tag{74}$$

**(2).** Next, the bounds of $g_2(\theta)$ is discussed. We continue to discuss the property of the parameter $\theta'$ between $\hat{\theta}$ and $\theta^*$.

Define $d_i(\theta^*, \theta') = \nabla\eta_{\theta'}^i - \nabla\eta_{\theta^*}^i$, since the first derivative function is also continuous. According to Assumption 3 and (51), we have $\|d_i(\theta^*, \theta')\| \leq 2b_2\|\theta' - \theta^*\|_2$.

For $g_2(\theta')$, we can further derive:

$$
\begin{aligned}
g_2(\theta') &\geq \frac{\zeta}{n}\sum_{i=1}^{n} \nabla\eta_{\theta'}^i (\nabla\eta_{\theta'}^i)^{\mathrm{T}} \\
&= \frac{\zeta}{n}\sum_{i=1}^{n} (\nabla\eta_{\theta^*}^i + d_i(\theta^*, \theta'))(\nabla\eta_{\theta^*}^i + d_i(\theta^*, \theta'))^{\mathrm{T}} \\
&\geq \frac{\zeta}{n}\sum_{i=1}^{n}\left( \nabla\eta_{\theta^*}^i(\nabla\eta_{\theta^*}^i)^{\mathrm{T}} + d_i(\theta^*, \theta')(\nabla\eta_{\theta^*}^i)^{\mathrm{T}} + \nabla\eta_{\theta^*}^i d_i(\theta^*, \theta')^{\mathrm{T}} \right).
\end{aligned}
\tag{75}
$$

Next, we combine $g_1(\theta')$ and $g_2(\theta')$ to discuss the property that $\theta'$ can satisfy. Define $X(\theta^*) = [\nabla\eta_{\theta^*}^1, \nabla\eta_{\theta^*}^2, ..., \nabla\eta_{\theta^*}^n]^{\mathrm{T}} \in \mathbb{R}^{n \times d}$ is the parameter matrix of the $n$ samples, then $\Sigma_D(\theta^*) = \frac{1}{n}X(\theta^*)^{\mathrm{T}}X(\theta^*)$.

Note that $|\nabla\eta_\theta^i| \leq 2b_1$, according to (45), we have

$$
\begin{aligned}
\nabla^2 l(\theta') &= g_1(\theta') + g_2(\theta') \\
&\geq \frac{\zeta}{n}X(\theta^*)^{\mathrm{T}}X(\theta^*) + \frac{\zeta}{n}\sum_{i=1}^{n} d_i(\theta^*, \theta')(\nabla\eta_{\theta^*}^i)^{\mathrm{T}} + \frac{\zeta}{n}\sum_{i=1}^{n}\nabla\eta_{\theta^*}^i d_i(\theta^*, \theta')^{\mathrm{T}} + g_1(\theta') \\
&\geq \frac{\zeta}{n}X(\theta^*)^{\mathrm{T}}X(\theta^*) - 8\zeta b_2 b_1\|\theta' - \theta^*\|_2 + g_1(\theta').
\end{aligned}
\tag{76}
$$

Combined with the previous derivation, we can summarize the following property about $\theta'$. For any $\theta' = \theta^* + \sigma\Delta, \sigma \in [0,1]$, the following event holds with the probability of $1 - \delta$.

$$\Delta^{\mathrm{T}}\nabla^2 l(\theta')\Delta \geq \zeta\|\Delta\|_{\Sigma_D(\theta^*)}^2 - \Delta^{\mathrm{T}}(8\zeta b_2 b_1\|\theta' - \theta^*\|_2)\Delta - \Delta^{\mathrm{T}}(b_1 c_1 + 2c_2)d\|\theta' - \theta^*\|_2\Delta - \Delta^{\mathrm{T}}dc_1\sqrt{\frac{8\log(2d^2/\delta)}{n}}\Delta. \tag{77}$$

Using Taylor expansion with Lagrangian remainder, we know that $\exists\theta' = \theta^* + e(\hat{\theta} - \theta^*), e \in [0,1], l(\theta^* + \Delta) - l(\theta^*) - \nabla l(\theta^*)^{\mathrm{T}}\Delta = \Delta^{\mathrm{T}}\nabla^2 l(\theta')\Delta$. Since (77) holds for all $\theta'$, we have

$$l(\theta^* + \Delta) - l(\theta^*) - \nabla l(\theta^*)^{\mathrm{T}}\Delta \geq \zeta\|\Delta\|_{\Sigma_D(\theta^*)}^2 - \Delta^{\mathrm{T}}(8\zeta b_1 b_2 + b_1 c_1 d + 2c_2 d)\|\Delta\|_2\Delta - dc_1\sqrt{\frac{8\log(2d^2/\delta)}{n}}\|\Delta\|_2^2.$$

Note that $\theta^*$ is optimal for $L(\theta)$, according to the realizability Assumption 5, we have

$$l(\theta^* + \Delta) - l(\theta^*) - \Delta^{\mathrm{T}}\nabla l(\theta^*) + \Delta^{\mathrm{T}}\nabla l(\theta^*) \leq 0. \tag{78}$$

Using the Cauchy-Schwarz inequality (Shah et al., 2016) for (78) and combining (77), we obtain

$$
\begin{aligned}
\|\Delta\|_{\Sigma_D(\theta^*)+\lambda I_d} \|\nabla l(\theta^*)\|_{(\Sigma_D(\theta^*)+\lambda I_d)^{-1}} &\geq -\Delta^{\mathrm{T}}\nabla l(\theta^*) \\
&\geq l(\theta^*+\Delta) - l(\theta^*) - \Delta^{\mathrm{T}}\nabla l(\theta^*) \\
&\geq \zeta\|\Delta\|_{\Sigma_D(\theta^*)}^2 - \Delta^{\mathrm{T}}(8\zeta b_1 b_2 + b_1 c_1 d + 2c_2 d)\|\Delta\|_2 \Delta \\
&\quad - dc_1\sqrt{\frac{8\log(2d^2/\delta)}{n}}\|\Delta\|_2^2.
\end{aligned}
\tag{79}
$$

The following goal is to transform (79) into an inequality equation about $\|\Delta\|$ while ensuring that the inequality still holds. Thus, we need to bound $\|\nabla l(\theta^*)\|_{(\Sigma_D(\theta^*)+\lambda I_d)^{-1}}$ further.

Recall

$$
\nabla l(\theta^*) = -\frac{1}{n}\sum_{i=1}^{n}\left[\left(p_i * \frac{1}{1+e^{\eta_{\theta*}^i}} - (1-p_i)\frac{e^{\eta_{\theta*}^i}}{1+e^{\eta_{\theta*}^i}}\right)\nabla\eta_{\theta*}^i\right].
$$

Since both $\nabla l(\theta^*)$ and $\Sigma_D(\theta^*)$ are related to $\theta^*$, following Zhu et al. (2023), we can use Lemma 2 to bound $\|\nabla l(\theta^*)\|_{(\Sigma_D(\theta^*)+\lambda I_d)^{-1}}$.

Recall (56)

$$
y_i = p_i * \frac{1}{1+e^{\eta_{\theta*}^i}} - (1-p_i)\frac{e^{\eta_{\theta*}^i}}{1+e^{\eta_{\theta*}^i}},
$$

and

$$
X(\theta^*) = [\nabla\eta_{\theta*}^1, \nabla\eta_{\theta*}^2, ..., \nabla\eta_{\theta*}^n]^{\mathrm{T}}.
$$

Define $Y = [x_1, ..., x_n]^{\mathrm{T}} \in \mathbb{R}^n$, with this notation, Rewrite

$$
\nabla l(\theta^*) = -\frac{1}{n}X(\theta^*)^{\mathrm{T}}Y,
$$

and

$$
\begin{aligned}
\|\nabla l(\theta^*)\|_{(\Sigma_D(\theta^*)+\lambda I_d)^{-1}} &= \sqrt{\nabla l(\theta^*)^{\mathrm{T}}(\Sigma_D(\theta^*)+\lambda I_d)^{-1}\nabla l(\theta^*)} \\
&= \sqrt{\frac{1}{n^2}Y^{\mathrm{T}}X(\theta^*)(\Sigma_D(\theta^*)+\lambda I_d)^{-1}X(\theta^*)^{\mathrm{T}}Y} \\
&= \sqrt{Y^{\mathrm{T}}MY},
\end{aligned}
$$

where $M := \frac{1}{n^2}X(\theta^*)(\Sigma_D(\theta^*)+\lambda I_d)^{-1}X(\theta^*)^{\mathrm{T}}$ is a n-dimensional square matrix. To determine the $\mathrm{tr}(M)$, $\mathrm{tr}(M^2)$ and $\|M\|_2$, we need to analyze some properties of $M$. First, perform singular value decomposition on $X(\theta^*)$:

$$
X(\theta^*) = U\Sigma V^{\mathrm{T}},
\tag{80}
$$

where $\Sigma \in \mathbb{R}^{n\times d}$, $U \in \mathbb{R}^{n\times n}$ and $V \in \mathbb{R}^{d\times d}$. $U$ and $V$ are two orthogonal matrices. The diagonal of $\Sigma$ contains $d$ singular values $\sigma_i$.

Calculate $(\Sigma_D(\theta^*)+\lambda I_d)^{-1}$ first

$$
\begin{aligned}
(\Sigma_D(\theta^*)+\lambda I_d)^{-1} &= (\frac{1}{n}V\Sigma^{\mathrm{T}}U^{\mathrm{T}}U\Sigma V^{\mathrm{T}} + \lambda I_d)^{-1} \\
&= n(V\Sigma^{\mathrm{T}}\Sigma V^{\mathrm{T}} + n\lambda I_d)^{-1} \quad \text{(Since } U \text{ is an orthogonal matrix.)} \\
&= n\left(V(\Lambda + n\lambda I_d)V^{\mathrm{T}}\right)^{-1} \\
&= nV(\Lambda + n\lambda I_d)^{-1}V^{\mathrm{T}}
\end{aligned}
$$

where $\Lambda = \Sigma^{\mathrm{T}}\Sigma \in \mathbb{R}^{d \times d}$ is a diagonal matrix with diagonal elements $\sigma_i^2$.

Then,

$$
\begin{aligned}
M &= \frac{1}{n^2}X(\theta^*)(\Sigma_D(\theta^*) + \lambda I_d)^{-1}X(\theta^*)^{\mathrm{T}} \\
&= \frac{1}{n}(U\Sigma V^{\mathrm{T}})(V(\Lambda + n\lambda I_d)^{-1}V^{\mathrm{T}})(V\Sigma^{\mathrm{T}}U^{\mathrm{T}}) \\
&= \frac{1}{n}U\Sigma(\Lambda + n\lambda I_d)^{-1}\Sigma^{\mathrm{T}}U^{\mathrm{T}} \quad \text{(Since } U \text{ and } V \text{ are orthogonal matrices.)}
\end{aligned}
$$

Since $\Lambda$ is a diagonal matrix, $(\Lambda + n\lambda I_d)^{-1}$ is also a $d$-dimensional diagonal matrix with diagonal elements $\frac{1}{n\lambda + \sigma_i^2}$. Furthermore, $\Sigma(\Lambda + n\lambda I_d)^{-1}\Sigma^{\mathrm{T}}$ is also a $d$-dimensional diagonal matrix with diagonal elements $\frac{\sigma_i^2}{n\lambda + \sigma_i^2}$. Thus, $M$ has only $d$ non-zero eigenvalues $\frac{\sigma_i^2}{n(n\lambda + \sigma_i^2)}$. We summarize the properties of $M$ as follows

$$\mathrm{Tr}(M) \le \frac{d}{n} \quad \text{(Since } \lambda > 0 \text{ and the trace of a matrix is equal to the sum of its eigenvalues.)} \tag{81}$$

$$\mathrm{Tr}\left(M^2\right) \le \frac{d}{n^2} \tag{82}$$

$$\|M\|_2 \le \frac{1}{n} \quad \text{(Since } \lambda > 0.) \tag{83}$$

Since $\mathbb{E}[y_i] = 0$ and $-1 \le y_i \le 1$, according to Hoeffding Lemma,

$$\mathbb{E}[e^{\kappa y_i}] \le e^{\frac{\kappa^2}{2}}$$

holds for any $\kappa$. Since $y_i$ are independent random variables,

$$\prod_{i=1}^{n} \mathbb{E}\left[e^{\kappa_i y_i}\right] = \mathbb{E}\left[e^{\sum_i^n \kappa_i y_i}\right]$$

holds for any $\kappa_1, \kappa_2, ..., \kappa_n$, which shows that random vector $Y$ satisfies (35).

According to Lemma 2, combined with (81)-(83), we have

$$\mathbb{P}\left[Y^{\mathrm{T}}MY > \frac{d}{n} + 2\sqrt{\frac{d}{n^2}t} + \frac{2t}{n}\right] \le e^{-t}.$$

Then, taking $\delta = e^{-t}$, i.e., $t = \log\frac{1}{\delta}$, with probability $1 - \delta$, we have

$$\|\nabla l(\theta^*)\|_{(\Sigma_D(\theta^*) + \lambda I_d)^{-1}} = \sqrt{Y^{\mathrm{T}}MY} \le \sqrt{\frac{d + 2\sqrt{d\log\frac{1}{\delta}} + 2\log\frac{1}{\delta}}{n}} \tag{84}$$

Using union bound, with probability at least $1 - \delta$, according to (79), we have

$$
\begin{aligned}
\sqrt{\frac{d + 2\sqrt{d\log\frac{2}{\delta}} + 2\log\frac{2}{\delta}}{n}}\|\Delta\|_{\Sigma_D(\theta^*) + \lambda I_d} &\ge \zeta\|\Delta\|_{\Sigma_D(\theta^*)}^2 - \Delta^{\mathrm{T}}(8\zeta b_1 b_2 + b_1 c_1 d + 2c_2 d)\|\Delta\|_2\Delta \\
&\quad - dc_1\sqrt{\frac{8\log(2d^2/\delta)}{n}}\|\Delta\|_2^2 \\
&= \zeta\|\Delta\|_{\Sigma_D(\theta^*) + \lambda I_d}^2 - (8\zeta b_1 b_2 + b_1 c_1 d + 2c_2 d)\|\Delta\|_2^3 \\
&\quad - (\zeta\lambda + dc_1\sqrt{\frac{8\log(2d^2/\delta)}{n}})\|\Delta\|_2^2 \\
&\ge \zeta\|\Delta\|_{\Sigma_D(\theta^*) + \lambda I_d}^2 - \underline{\beta}^3(8\zeta b_1 b_2 + b_1 c_1 d + 2c_2 d)\|\Delta\|_{\Sigma_D(\theta^*) + \lambda I_d}^3 \\
&\quad - \underline{\beta}^2(\zeta\lambda + dc_1\sqrt{\frac{8\log(2d^2/\delta)}{n}})\|\Delta\|_{\Sigma_D(\theta^*) + \lambda I_d}^2.
\end{aligned}
$$

The third inequality is due to the consistency of the norm, i.e.,

$$\|v\|_2^2 \le \underline{\beta}^2 \|v\|_{\Sigma_D(\theta^*)+\lambda I_d}^2 \tag{85}$$

holds, for any $v \in \mathbb{R}^d$, since $\frac{1}{\underline{\beta}^2}$ is the smallest eigenvalue of $\Sigma_D(\theta^*) + \lambda I_d$.

The above inequalities are organized into

$$-a_2\|\Delta\|_{\Sigma_D(\theta^*)+\lambda I_d}^2 + a_1\|\Delta\|_{\Sigma_D(\theta^*)+\lambda I_d} - a_0 \le 0, \tag{86}$$

where

$$a_2 = \underline{\beta}^3(8\zeta b_1 b_2 + b_1 c_1 d + 2 c_2 d) \ge 0$$

$$a_1 = \zeta - \underline{\beta}^2(\zeta\lambda + dc_1\sqrt{\frac{8\log(2d^2/\delta)}{n}}), \text{ and}$$

$$a_0 = \sqrt{\frac{d + 2\sqrt{d\log\frac{2}{\delta}} + 2\log\frac{2}{\delta}}{n}}.$$

Next, we need to solve the inequality (86). If $c_1 = 0$, by definition, this implies that $r_\theta$ is linear, and all its second-order derivatives are 0. Consequently, we have $b2 = c2 = 0$, and $a2 = 0$. Therefore,

$$\|\Delta\|_{\Sigma_D(\theta^*)+\lambda I_d} \le \frac{a_0}{a_1}. \tag{87}$$

If $c_1 > 0$, since $a_1^2 - 4 * a_0 * a_2$ is a monotonically increasing function of $n$, and $\lim_{n\to\infty} a_1^2 - 4 * a_0 * a_2 \to \zeta(1 - \underline{\beta}^2\lambda) > 0$, there exists $n^*$, such that when $n > n^*$, $a_1^2 - 4 * a_0 * a_2 > 0$ we have

$$\|\Delta\|_{\Sigma_D(\theta^*)+\lambda I_d} \le \frac{a_1 - \sqrt{a_1^2 - 4 * a_0 * a_2}}{2a_2}$$

$$or \|\Delta\|_{\Sigma_D(\theta^*)+\lambda I_d} \ge \frac{a_1 + \sqrt{a_1^2 - 4 * a_0 * a_2}}{2a_2}. \tag{88}$$

Since $B < \frac{a_1 + \sqrt{a_1^2 - 4*a_0*a_2}}{4a_2\sqrt{4b_1^2 + \lambda}}$, we can conclude that

$$\|\Delta\|_{\Sigma_D(\theta^*)+\lambda I_d} \le \frac{a_1 - \sqrt{a_1^2 - 4 * a_0 * a_2}}{2a_2}. \tag{89}$$

This completes the proof. $\square$

## C.2 Proof of the Theorem 2

**Theorem 2** (Restatement). *Under the Assumption 2-5, if $\underline{\hat\alpha} > 0$, for any $\lambda \ge 0$, with probability $1 - \delta$, (1) When $b_2 = 0$, we have*

$$\|\Delta\|_{\Sigma_D(\hat\theta)+\lambda I_d} \le \frac{\hat a_0}{\hat a_1}. \tag{90}$$

*(2) When $b_2 > 0$, $\exists n^*$, after $n > n^*$, we have*

$$\|\Delta\|_{\Sigma_D(\hat\theta)+\lambda I_d} \le \frac{\hat a_1 - \sqrt{\hat a_1^2 - 4 * \hat a_0 * \hat a_2}}{2\hat a_2}, \tag{91}$$

*if* $B < \frac{\hat{a}_1 + \sqrt{\hat{a}_1^2 - 4*\hat{a}_0*\hat{a}_2}}{4\hat{a}_2 \sqrt{4b_1^2 + \lambda}}$, *where*

$$\hat{a}_2 = \underline{\hat{\beta}}^3 (8\zeta b_1 b_2 + b_1 c_1 d + 2c_2 d) \geq 0,$$

$$\hat{a}_1 = \zeta - \underline{\hat{\beta}}^2 (\zeta\lambda + dc_1 \sqrt{\frac{8\log(2d^2/\delta)}{n}}),$$

$$\hat{a}_0 = \underline{\hat{\beta}} b_1 \sqrt{\frac{8d\log(2d/\delta)}{n}}.$$

$\underline{\hat{\beta}} = \frac{1}{\sqrt{\lambda + \underline{\hat{\alpha}}}}$, *and* $\underline{\hat{\alpha}}$ *is the minimum eigenvalue of* $\Sigma_D(\hat{\theta})$, *with* $\Sigma_D(\hat{\theta}) = \frac{1}{n} \sum_{i=1}^{n} \nabla\eta_{\hat{\theta}}^i (\nabla\eta_{\hat{\theta}}^i)^{\mathrm{T}}$.

*Proof.* Note that $\nabla l(\theta) \in \mathbb{R}^d$, then

$$\nabla l(\theta) = -\frac{1}{n} \left[ \sum_{i=1}^{n} \left( p_i * \frac{e^{-\eta_\theta^i} * \nabla\eta_\theta^i}{1 + e^{-\eta_\theta^i}} - (1 - p_i) \frac{e^{\eta_\theta^i} * \nabla\eta_\theta^i}{1 + e^{\eta_\theta^i}} \right) \right]$$

$$= -\frac{1}{n} \left[ \sum_{i=1}^{n} \left( p_i * \frac{e^{-\eta_\theta^i}}{1 + e^{-\eta_\theta^i}} - (1 - p_i) \frac{e^{\eta_\theta^i}}{1 + e^{\eta_\theta^i}} \right) \nabla\eta_\theta^i \right]$$

$$= -\frac{1}{n} \left[ \sum_{i=1}^{n} \left( p_i * \frac{1}{1 + e^{\eta_\theta^i}} - (1 - p_i) \frac{e^{\eta_\theta^i}}{1 + e^{\eta_\theta^i}} \right) \nabla\eta_\theta^i \right]. \tag{92}$$

The next step is to compute the Hessian matrix, $\nabla^2 l(\theta) \in \mathbb{R}^{d \times d}$:

$$\nabla^2 l(\theta) = -\frac{1}{n} \left[ \sum_{i=1}^{n} \left( p_i * \frac{1}{1 + e^{\eta_\theta^i}} - (1 - p_i) \frac{e^{\eta_\theta^i}}{1 + e^{\eta_\theta^i}} \right) \nabla^2 \eta_\theta^i \right]$$

$$- \frac{1}{n} \left[ \sum_{i=1}^{n} \left( p_i * \frac{-e^{\eta_\theta^i}}{(1 + e^{\eta_\theta^i})^2} - (1 - p_i) \frac{e^{\eta_\theta^i}}{1 + e^{\eta_\theta^i}} + (1 - p_i) \frac{(e^{\eta_\theta^i})^2}{(1 + e^{\eta_\theta^i})^2} \right) \nabla\eta_\theta^i (\nabla\eta_\theta^i)^{\mathrm{T}} \right]$$

$$= g_1(\theta) + g_2(\theta). \tag{93}$$

where

$$g_1(\theta) = -\frac{1}{n} \sum_{i=1}^{n} \left[ \left( p_i * \frac{1}{1 + e^{\eta_\theta^i}} - (1 - p_i) \frac{e^{\eta_\theta^i}}{1 + e^{\eta_\theta^i}} \right) \nabla^2 \eta_\theta^i \right]. \tag{94}$$

and

$$g_2(\theta) = \frac{1}{n} \sum_{i=1}^{n} \left( \frac{e^{\eta_\theta^i}}{(1 + e^{\eta_\theta^i})^2} \nabla\eta_\theta^i (\nabla\eta_\theta^i)^{\mathrm{T}} \right). \tag{95}$$

Then, we need to bound these two items respectively.

For convenience, noting that in the subsequent analysis we use the variable $\eta_\theta^i = r_\theta(\tau_1^i) - r_\theta(\tau_2^i)$, we first convert the constraints on the reward function in Assumption 3 into constraints on $\eta_\theta^i$. For any $\theta \in \Theta$, $\tau_1, \tau_2 \in \mathcal{T}$, the following inequalities are hold according to the triangle inequality of norms (Horn & Johnson,

2012):

$$|\eta_\theta(\tau_1,\tau_2)| = |r_\theta(\tau_1) - r_\theta(\tau_2)| \le |r_\theta(\tau_1)| + |r_\theta(\tau_2)| \le 2b_0 \tag{96}$$

$$\|\nabla\eta_\theta(\tau_1,\tau_2)\|_2 = \|\nabla r_\theta(\tau_1) - \nabla r_\theta(\tau_2)\|_2 \le \|\nabla r_\theta(\tau_1)\|_2 + \|\nabla r_\theta(\tau_2)\|_2 \le 2b_1 \tag{97}$$

$$\|\nabla^2\eta_\theta(\tau_1,\tau_2)\|_2 = \|\nabla^2 r_\theta(\tau_1) - \nabla^2 r_\theta(\tau_2)\|_2 \le \|\nabla^2 r_\theta(\tau_1)\|_2 + \|\nabla^2 r_\theta(\tau_2)\|_2 \le 2b_2 \tag{98}$$

$$|\frac{\partial^2\eta_\theta(\tau_1,\tau_2)}{\theta_i\theta_j}| = |\frac{\partial^2 r_\theta(\tau_1)}{\theta_i\theta_j} - \frac{\partial^2 r_\theta(\tau_2)}{\theta_i\theta_j}| \le 2c_1 \tag{99}$$

$$\|\nabla\frac{\partial^2\eta_\theta(\tau_1,\tau_2)}{\theta_i\theta_j}\|_2 = \|\nabla\frac{\partial^2 r_\theta(\tau_1)}{\theta_i\theta_j} - \nabla\frac{\partial^2 r_\theta(\tau_2)}{\theta_i\theta_j}\|_2 \le 2c_2, \tag{100}$$

Next, we will discuss the boundaries of $g_1(\theta)$ and $g_2(\theta)$, respectively.

**(1).** For $g_1(\theta)$, we discuss the bound of $g_1(\theta')$ in the range of $\theta' = \theta^* + \sigma\Delta, \sigma \in [0,1]$. Recall that $\Delta = \hat{\theta} - \theta^*$. Intuitively, we leverage the continuity of $r_\theta \in \mathcal{R}$ with respect to $\theta \in \Theta$ to explore the property that $\theta'$ satisfies, where $\theta'$ lies between $\theta^*$ and $\hat{\theta}$.

Then,

$$g_1(\theta') = -\frac{1}{n}\sum_{i=1}^n \left(p_i * \frac{1}{1+e^{\eta_{\theta'}^i}} - (1-p_i)\frac{e^{\eta_{\theta'}^i}}{1+e^{\eta_{\theta'}^i}}\right)\nabla^2\eta_{\theta'}^i.$$

Next, we aim to bound matrix $g_1(\theta') \in \mathbb{R}^{d\times d}$ using Lemma 3, and therefore perform an element-wise analysis of $g_1(\theta')$ first.

In order to reduce the pressure of symbolic expression, we define the elements of the matrix $\nabla^2\eta_{\theta'}^i$ as $a'^i_{jk}$, where $i \le n$, and $1 \le j,k \le d$. The apostrophe indicates that this element is associated with $\theta'$. From (100), we get $-2c_1 \le a'^i_{jk} \le 2c_1$.

Then, the elements of the matrix $g_1(\theta')$ can be expressed as

$$g'_{jk} = -\frac{1}{n}\sum_{i=1}^n g'^i_{jk}, \tag{101}$$

where $g'^i_{jk} = \left(p_i * \frac{1}{1+e^{\eta_{\theta'}^i}} - (1-p_i)\frac{e^{\eta_{\theta'}^i}}{1+e^{\eta_{\theta'}^i}}\right)a'^i_{jk}$. Decompose $g'_{jk}$ into two parts as shown in (102): $g'_{jk} - g^*_{jk}$ and $g^*_{jk}$. The first term is associated with $\theta' - \theta^*$, while the other involves $\theta^*$ and is expected to be 0.

$$g'_{jk} = \left(g'_{jk} - g^*_{jk}\right) + g^*_{jk}, \tag{102}$$

where $g^*_{jk} = -\frac{1}{n}\sum_{i=1}^n g^{*i}_{jk}$, and

$$g^{*i}_{jk} = \left(p_i * \frac{1}{1+e^{\eta_{\theta*}^i}} - (1-p_i)\frac{e^{\eta_{\theta*}^i}}{1+e^{\eta_{\theta*}^i}}\right)a^{*i}_{jk}. \tag{103}$$

Next, we need to find bounds for $\left(g'_{jk} - g^*_{jk}\right)$ and $g^*_{jk}$ respectively.

We will begin by finding the bound for $g^*_{jk}$. After fixing the samples $\tau_1^i$ and $\tau_2^i$, $g^{*i}_{jk}$ is a random variable, with its randomness arising from $p_i$. Define an auxiliary random variable

$$y_i = p_i * \frac{1}{1+e^{\eta_{\theta*}^i}} - (1-p_i)\frac{e^{\eta_{\theta*}^i}}{1+e^{\eta_{\theta*}^i}}. \tag{104}$$

Then, take the expectation on both sides of equation (104), we have

$$\mathbb{E}[y_i] = \mathbb{E}\left[p_i * \frac{1}{1 + e^{\eta_{\theta*}^i}}\right] - \mathbb{E}\left[(1 - p_i)\frac{e^{\eta_{\theta*}^i}}{1 + e^{\eta_{\theta*}^i}}\right]. \tag{105}$$

Note that the randomness on the right side of the equation (105) comes from $p_i$, and $\frac{e^{\eta_{\theta*}^i}}{1+e^{\eta_{\theta*}^i}}$ and $\frac{1}{1+e^{\eta_{\theta*}^i}}$ are constants. We can further obtain

$$\mathbb{E}[y_i] = \mathbb{E}[p_i] * \frac{1}{1 + e^{\eta_{\theta*}^i}} - (1 - \mathbb{E}[p_i])\frac{e^{\eta_{\theta*}^i}}{1 + e^{\eta_{\theta*}^i}} = 0. \tag{106}$$

Similarly, since $a^{*i}_{jk}$ is independent of $p_i$,

$$\mathbb{E}[g^{*i}_{jk}] = \mathbb{E}[y_i]a^{*i}_{jk} = 0 \tag{107}$$

Since $|y_i| <= 1$, $g^{*i}_{jk}$ is also a bounded random variable, with $-2c_1 \leq g^{*i}_{jk} \leq 2c_1$. Thus, according to Hoeffding's inequality, we have

$$\mathbb{P}(|g^*_{jk}| \geq t) = \mathbb{P}\left(\left|\frac{1}{n}\sum_{i=1}^{n} g^{*i}_{jk} - \mathbb{E}[\frac{1}{n}\sum_{i=1}^{n} g^{*i}_{jk}]\right| \geq t\right) \leq 2e^{\left(-\sum_{i=1}^{n}\frac{2n^2 t^2}{(4c_1)^2}\right)}. \tag{108}$$

Simplify (108) and get

$$\mathbb{P}(|g^*_{jk}| \geq t) \leq 2e^{\frac{-nt^2}{8c_1^2}}. \tag{109}$$

Rewriting (109) gives

$$\mathbb{P}(|g^*_{jk}| \leq t) \geq 1 - 2e^{\frac{-nt^2}{8c_1^2}}. \tag{110}$$

Now, let

$$\delta = 2e^{\frac{-nt^2}{8c_1^2}}, \tag{111}$$

which leads to

$$t = c_1\sqrt{\frac{8\log(2/\delta)}{n}}. \tag{112}$$

Thus, $|g^*_{jk}|$ can be bounded with the probability of $1 - \delta$

$$|g^*_{jk}| \leq c_1\sqrt{\frac{8\log(2/\delta)}{n}}, \tag{113}$$

for some special $j, k$.

Then, consider the bound of another part, $g'_{jk} - g^*_{jk}$. The idea behind proving its boundedness is to use the Lipschitz continuity condition to transform it into a bound related to the estimation error $\theta^* - \theta'$.

$$g'_{jk} - g^*_{jk} = -\frac{1}{n}\sum_{i=1}^{n}\left(p_i * \frac{1}{1 + e^{\eta_{\theta'}^i}} - (1 - p_i)\frac{e^{\eta_{\theta'}^i}}{1 + e^{\eta_{\theta'}^i}}\right)a'^i_{jk} + \frac{1}{n}\sum_{i=1}^{n}\left(p_i * \frac{1}{1 + e^{\eta_{\theta*}^i}} - (1 - p_i)\frac{e^{\eta_{\theta*}^i}}{1 + e^{\eta_{\theta*}^i}}\right)a^{*i}_{jk},$$

Consider an auxiliary function

$$f_{jk}^i(\theta) = \left( p_i * \frac{1}{1 + e^{\eta_\theta^i}} - (1 - p_i) \frac{e^{\eta_\theta^i}}{1 + e^{\eta_\theta^i}} \right) \frac{\partial^2 \eta_\theta(\tau)}{\theta_j \theta_k} \tag{114}$$
$$= f_1^i(\theta) f_2^i(\theta).$$

Note that

$$\nabla f_1^i(\theta) = \frac{\partial f_1^i(\theta)}{\partial \theta} = p_i * \frac{-e^{\eta_\theta^i}}{(1 + e^{\eta_\theta^i})^2} * \nabla \eta_\theta(\tau_1^i, \tau_2^i) - (1 - p_i) \frac{e^{\eta_\theta^i}}{(1 + e^{\eta_\theta^i})^2} * \nabla \eta_\theta(\tau_1^i, \tau_2^i), \tag{115}$$

according to (98), we have

$$\|\nabla f_1^i(\theta)\|_2 \le 0.5 b_1, \tag{116}$$

since $|\frac{-e^x}{(1+e^x)^2}| \le 0.25$ and $|\frac{e^x}{(1+e^x)^2}| \le 0.25$ always holds for any $x$.

Thus,

$$\nabla f_{jk}^i(\theta) = \nabla f_1^i(\theta) f_2^i(\theta) + f_1^i(\theta) \nabla f_2^i(\theta). \tag{117}$$

Furthermore, we have

$$\|\nabla f_{jk}^i(\theta)\|_2 \le \|\nabla f_1^i(\theta) f_2^i(\theta)\|_2 + \|f_1^i(\theta) \nabla f_2^i(\theta)\|_2 \le b_1 c_1 + 2 c_2, \tag{118}$$

since $|\frac{1}{1+e^x}| \le 1$ and $|\frac{e^x}{1+e^x}| \le 1$ always holds for any $x > 0$.

We further know that $f_{jk}^i(\theta)$ is continuous with respect to $\theta$, since $f_1^i(\theta)$ and $f_2^i(\theta)$ is continuous according to the assumptions. According to the mean value theorem, we have

$$f^i(\theta') - f^i(\theta^*) = \nabla f^i(\bar{\theta})^{\mathrm{T}}(\theta' - \theta^*)$$

holds for some $\bar{\theta}$. Thus,

$$|f^i(\theta') - f^i(\theta^*)| = |\nabla f^i(\bar{\theta})^{\mathrm{T}}(\theta' - \theta^*)| \le \|\nabla f^i(\bar{\theta})\|_2 \|\theta' - \theta^*\|_2 \le (b_1 c_1 + 2 c_2)\|\theta' - \theta^*\|_2$$

holds for any $1 \le i \le n$, and $1 \le j, k \le d$. Combined with the above analysis, we have

$$\begin{aligned} \left| g_{jk}' - g_{jk}^* \right| &= \left| \frac{1}{n} \sum_{i=1}^n \left( f_{jk}^i(\theta^*) - f_{jk}^i(\theta') \right) \right| \\ &\le \frac{1}{n} \sum_{i=1}^n |f_{jk}^i(\theta^*) - f_{jk}^i(\theta')| \\ &\le (b_1 c_1 + 2 c_2)\|\theta' - \theta^*\|_2 \end{aligned}$$

According to (102) and (113),

$$|g_{jk}'| \le (b_1 c_1 + 2 c_2)\|\theta' - \theta^*\|_2 + c_1 \sqrt{\frac{8 \log(2/\delta)}{n}} \tag{119}$$

holds with the probability of $1 - \delta$, for special $j, k$. Using union bound, with probability $1 - \delta$, we have

$$|g_{jk}'| \le (b_1 c_1 + 2 c_2)\|\theta' - \theta^*\|_2 + c_1 \sqrt{\frac{8 \log(2 d^2/\delta)}{n}}, \tag{120}$$

for any $j, k$.

According to (120) and Lemma 3, We get the following results about the spectral norm of matrix $g_1(\theta')$. For any $\theta' = \theta^* + \sigma\Delta, \sigma \in [0,1]$, with the probability of $1 - \delta$, we have

$$g_1(\theta') \leq \left(d(b_1c_1 + 2c_2)\|\theta' - \theta^*\|_2 + dc_1\sqrt{\frac{8\log(2d^2/\delta)}{n}}\right)I_d. \tag{121}$$

**(2).** Next, the bounds of $g_2(\theta)$ is discussed. We continue to discuss the property of the parameter $\theta'$ between $\hat{\theta}$ and $\theta^*$. The key difference from the proof of Theorem 1 is that this time we expands around $\theta'$. Define $d_i(\hat{\theta}, \theta') = \nabla\eta_{\theta'}^i - \nabla\eta_{\hat{\theta}}^i$, since the first derivative function is also continuous according to Assumption 3 and (51), we have $\|d_i(\hat{\theta}, \theta')\| \leq 2b_2\|\theta' - \hat{\theta}\|_2$. Then, $g_2(\theta')$ can further obtain as follows.

$$\begin{aligned}
g_2(\theta') &\geq \frac{\zeta}{n}\sum_{i=1}^n \nabla\eta_{\theta'}^i(\nabla\eta_{\theta'}^i)^{\mathrm{T}} \\
&= \frac{\zeta}{n}\sum_{i=1}^n (\nabla\eta_{\hat{\theta}}^i + d_i(\hat{\theta}, \theta'))(\nabla\eta_{\hat{\theta}}^i + d_i(\hat{\theta}, \theta'))^{\mathrm{T}} \\
&\geq \frac{\zeta}{n}\sum_{i=1}^n \left(\nabla\eta_{\hat{\theta}}^i(\nabla\eta_{\hat{\theta}}^i)^{\mathrm{T}} + d_i(\hat{\theta}, \theta')(\nabla\eta_{\hat{\theta}}^i)^{\mathrm{T}} + \nabla\eta_{\hat{\theta}}^i d_i(\hat{\theta}, \theta')^{\mathrm{T}}\right).
\end{aligned} \tag{122}$$

In the following, we combine $g_1(\theta')$ and $g_2(\theta')$ to discuss the property that $\theta'$ can satisfy. Define $X(\hat{\theta}) = [\nabla\eta_{\hat{\theta}}^1, \nabla\eta_{\hat{\theta}}^2, ..., \nabla\eta_{\hat{\theta}}^n]^{\mathrm{T}} \in \mathbb{R}^{n \times d}$ is the parameter matrix of the $n$ samples, then $\Sigma_D(\hat{\theta}) = \frac{1}{n}X(\hat{\theta})^{\mathrm{T}}X(\hat{\theta})$. For now, Note that $|\nabla\eta_{\hat{\theta}}^i| \leq 2b_1$, we have

$$\begin{aligned}
\nabla^2 l(\theta') &= g_1(\theta') + g_2(\theta') \\
&\geq \frac{\zeta}{n}X(\hat{\theta})^{\mathrm{T}}X(\hat{\theta}) + \frac{\zeta}{n}\sum_{i=1}^n d_i(\hat{\theta}, \theta')(\nabla\eta_{\hat{\theta}}^i)^{\mathrm{T}} + \frac{\zeta}{n}\sum_{i=1}^n \nabla\eta_{\hat{\theta}}^i d_i(\hat{\theta}, \theta')^{\mathrm{T}} + g_1(\theta') \\
&\geq \frac{\zeta}{n}X(\hat{\theta})^{\mathrm{T}}X(\hat{\theta}) - 8\zeta b_2 b_1\|\theta' - \hat{\theta}\|_2 + g_1(\theta').
\end{aligned} \tag{123}$$

Combined with the previous derivation, we can summarize the following property about $\theta'$. For any $\theta' = \theta^* + \sigma\Delta, \sigma \in [0,1]$, with the probability of $1 - \delta$, the following event is established.

$$\Delta^{\mathrm{T}}\nabla^2 l(\theta')\Delta \geq \zeta\|\Delta\|_{\Sigma_D(\hat{\theta})}^2 - \Delta^{\mathrm{T}}(8\zeta b_2 b_1\|\theta' - \hat{\theta}\|_2)\Delta - \Delta^{\mathrm{T}}d(b_1c_1 + 2c_2)\|\theta' - \theta^*\|_2\Delta - \Delta^{\mathrm{T}}dc_1\sqrt{\frac{8\log(2d^2/\delta)}{n}}\Delta. \tag{124}$$

Using Taylor expansion with Lagrangian remainder, we know that $\exists\theta' = \theta^* + e(\hat{\theta} - \theta^*), e \in [0,1], l(\theta^* + \Delta) - l(\theta^*) - \nabla l(\theta^*)^{\mathrm{T}}\Delta = \Delta^{\mathrm{T}}\nabla^2 l(\theta')\Delta$. Since (124) holds for all $\theta'$, we have

$$l(\theta^* + \Delta) - l(\theta^*) - \nabla l(\theta^*)^{\mathrm{T}}\Delta \geq \zeta\|\Delta\|_{\Sigma_D(\hat{\theta})}^2 - \Delta^{\mathrm{T}}(8\zeta b_1 b_2 + b_1 c_1 d + 2c_2 d)\|\Delta\|_2\Delta - dc_1\sqrt{\frac{8\log(2d^2/\delta)}{n}}\|\Delta\|_2^2.$$

Note that $\theta^*$ is optimal for $L(\theta)$, according to the realizability Assumption 5, we have

$$l(\theta^* + \Delta) - l(\theta^*) - \Delta^{\mathrm{T}}\nabla l(\theta^*) + \Delta^{\mathrm{T}}\nabla l(\theta^*) \leq 0. \tag{125}$$

Using the Cauchy-Schwarz inequality for (125) and combining (124), we obtain

$$\begin{aligned}
\|\Delta\|_2\|\nabla l(\theta^*)\|_2 &\geq -\Delta^{\mathrm{T}}\nabla l(\theta^*) \\
&\geq l(\theta^* + \Delta) - l(\theta^*) - \Delta^{\mathrm{T}}\nabla l(\theta^*) \\
&\geq \zeta\|\Delta\|_{\Sigma_D(\hat{\theta})}^2 - \Delta^{\mathrm{T}}(8\zeta b_1 b_2 + b_1 c_1 d + 2c_2 d)\|\Delta\|_2\Delta - dc_1\sqrt{\frac{8\log(2d^2/\delta)}{n}}\|\Delta\|_2^2.
\end{aligned} \tag{126}$$

The following goal is to transform (126) into an inequality equation about $\|\Delta\|$ while ensuring that the inequality still holds. Thus, we need to bound $\|\nabla l(\theta^*)\|_2$ further. Since $\nabla l(\theta^*)$ is related to $\theta^*$ but $\Sigma_D(\hat{\theta})$ is related to $\hat{\theta}$, Lemma 2 does not apply. Fortunately, we can obtain an acceptable result by controlling the bounds of the elements of $\nabla l(\theta^*)$ and applying the union bound.

Recall

$$\nabla l(\theta^*) = -\frac{1}{n}\left[\sum_{i=1}^{n}\left(p_i * \frac{1}{1+e^{\eta_{\theta^*}^i}} - (1-p_i)\frac{e^{\eta_{\theta^*}^i}}{1+e^{\eta_{\theta^*}^i}}\right)\nabla\eta_{\theta^*}^i\right],$$

and $\nabla\eta_{\theta^*}^i \in \mathbb{R}^d$ is a bounded vector after fixing $\tau_1^i$ and $\tau_2^i$.

With a slight abuse of notation, let the elements of $\nabla\eta_{\theta^*}^i$ be $a_j^i (1 \le j \le d, 1 \le i \le n)$. According to (104), we obtain the elements $a_j (1 \le j \le d)$ of $\nabla l(\theta^*)$ as follows.

$$a_j = -\frac{1}{n}\sum_{i=1}^{n}y_i a_j^i. \tag{127}$$

Since the randomness of $y_i$ comes from $p_i$ and is independent of $a_j^i$, based on (106), we have

$$\mathbb{E}[y_i a_j^i] = \mathbb{E}[y_i]a_j^i = 0. \tag{128}$$

Since $\|\nabla\eta_{\theta^*}^i\|_2 \le 2b_1$, and $a_j^i$ is the element of $\eta_{\theta^*}^i$, $|a_j^i| \le 2b_1$ always holds.

Combining $|y_i| \le 1$, $y_i a_j^i$ is a bounded random variable with $-2b_1 \le y_i a_j^i \le 2_1$.

Thus, according to Hoeffding's inequality, we have

$$\mathbb{P}(|a_j| \ge t) = \mathbb{P}\left(\left|\frac{1}{n}\sum_{i=1}^{n}y_i a_j^i - \mathbb{E}[\frac{1}{n}\sum_{i=1}^{n}y_i a_j^i]\right| \ge t\right) \le 2e^{\left(-\frac{2n^2t^2}{\sum_{i=1}^{n}(4b_1)^2}\right)}. \tag{129}$$

Simplify (129) and get

$$\mathbb{P}(|a_j| \ge t) \le 2e^{\frac{-nt^2}{8b_1^2}}. \tag{130}$$

Rewriting (130) gives

$$\mathbb{P}(|a_j| \le t) \ge 1 - 2e^{\frac{-nt^2}{8b_1^2}}. \tag{131}$$

Now, let

$$\delta = 2e^{\frac{-nt^2}{8b_1^2}}, \tag{132}$$

which leads to

$$t = b_1\sqrt{\frac{8\log(2/\delta)}{n}} \tag{133}$$

Thus, $|a_j|$ can be bounded with the probability of $1-\delta$

$$|a_j| \le b_1\sqrt{\frac{8\log(2/\delta)}{n}}, \tag{134}$$

for some special $j$. Using union bound, with at least probability $1-\delta$, we have

$$|a_j| \le b_1\sqrt{\frac{8\log(2d/\delta)}{n}}, \tag{135}$$

for any $j$. Therefore, we arrive at the final conclusion that

$$\|\nabla l(\theta^*)\|_2 \leq b_1 \sqrt{\frac{8d\log(2d/\delta)}{n}} \tag{136}$$

holds with a probability of at least $1 - \delta$.

Next, we use the compatibility of norms to unify the norm forms on both sides of the inequality (126). According to the compatibility of norms, we have

$$\|v\|_2 \leq \underline{\hat{\beta}}\|v\|_{\Sigma_D(\hat{\theta})+\lambda I_d}$$

holds for any $v \in \mathbb{R}^d$, since $\underline{\hat{\beta}} = \frac{1}{\sqrt{\lambda+\underline{\hat{\alpha}}}}$, and $\underline{\hat{\alpha}}$ is the minimum eigenvalue of $\Sigma_D(\hat{\theta})$ and $\frac{1}{\underline{\hat{\beta}}}$ is the smallest eigenvalue of $\Sigma_D(\hat{\theta}) + \lambda I_d$.

Using union bound, with probability at least $1 - \delta$, we have

$$\underline{\hat{\beta}}b_1\sqrt{\frac{8d\log(2d/\delta)}{n}}\|\Delta\|_{\Sigma_D(\hat{\theta})+\lambda I_d} \geq \zeta\|\Delta\|^2_{\Sigma_D(\hat{\theta})} - (8\zeta b_1 b_2 + b_1 c_1 d + 2c_2 d)\|\Delta\|^3_2 - dc_1\sqrt{\frac{8\log(2d^2/\delta)}{n}}\|\Delta\|^2_2$$

$$= \zeta\|\Delta\|^2_{\Sigma_D(\hat{\theta})+\lambda I_d} - (8\zeta b_1 b_2 + b_1 c_1 d + 2c_2 d)\|\Delta\|^3_2$$

$$- (\zeta\lambda + dc_1\sqrt{\frac{8\log(2d^2/\delta)}{n}})\|\Delta\|^2_2$$

$$\geq \zeta\|\Delta\|^2_{\Sigma_D(\hat{\theta})+\lambda I_d} - \underline{\hat{\beta}}^3(8\zeta b_1 b_2 + b_1 c_1 d + 2c_2 d)\|\Delta\|^3_{\Sigma_D(\hat{\theta})+\lambda I_d}$$

$$- \underline{\hat{\beta}}^2(\zeta\lambda + dc_1\sqrt{\frac{8\log(2d^2/\delta)}{n}})\|\Delta\|^2_{\Sigma_D(\hat{\theta})+\lambda I_d}.$$

The above inequalities are organized into

$$-\hat{a}_2\|\Delta\|^2_{\Sigma_D(\hat{\theta})+\lambda I_d} + \hat{a}_1\|\Delta\|_{\Sigma_D(\hat{\theta})+\lambda I_d} - \hat{a}_0 \leq 0, \tag{137}$$

where

$$\hat{a}_2 = \underline{\hat{\beta}}^3(8\zeta b_1 b_2 + b_1 c_1 d + 2c_2 d) \geq 0,$$

$$\hat{a}_1 = \zeta - \underline{\hat{\beta}}^2(\zeta\lambda + dc_1\sqrt{\frac{8\log(2d^2/\delta)}{n}}), \text{ and}$$

$$\hat{a}_0 = \underline{\hat{\beta}}b_1\sqrt{\frac{8d\log(2d/\delta)}{n}}.$$

Next, we need to solve the inequality (137). If $c_1 = 0$, by definition, this implies that $r_\theta$ is linear, and all its second-order derivatives are 0. Consequently, we have $b2 = c2 = 0$, and $a2 = 0$. Therefore,

$$\|\Delta\|_{\Sigma_D(\hat{\theta})+\lambda I_d} \leq \frac{\hat{a}_0}{\hat{a}_1}.$$

If $c_1 > 0$, since $\hat{a}_1^2 - 4 * \hat{a}_0 * \hat{a}_2$ is a monotonically increasing function of $n$, and $\lim_{n\to\infty} \hat{a}_1^2 - 4 * \hat{a}_0 * \hat{a}_2 \to \zeta(1 - \underline{\hat{\beta}}^2\lambda) > 0$, there exists $n^*$, such that when $n > n^*$, $\hat{a}_1^2 - 4 * \hat{a}_0 * \hat{a}_2 > 0$ we have

$$\|\Delta\|_{\Sigma_D(\hat{\theta})+\lambda I_d} \leq \frac{\hat{a}_1 - \sqrt{\hat{a}_1^2 - 4 * \hat{a}_0 * \hat{a}_2}}{2\hat{a}_2}$$

$$or\|\Delta\|_{\Sigma_D(\hat{\theta})+\lambda I_d} \geq \frac{\hat{a}_1 + \sqrt{\hat{a}_1^2 - 4 * \hat{a}_0 * \hat{a}_2}}{2\hat{a}_2}. \tag{138}$$

Since $B < \frac{\hat{a}_1 + \sqrt{\hat{a}_1^2 - 4*\hat{a}_0*\hat{a}_2}}{4\hat{a}_2\sqrt{4b_1^2+\lambda}}$, we can conclude that

$$\|\Delta\|_{\Sigma_D(\hat{\theta})+\lambda I_d} \leq \frac{\hat{a}_1 - \sqrt{\hat{a}_1^2 - 4*\hat{a}_0*\hat{a}_2}}{2\hat{a}_2}.$$

This completes the proof. $\qquad\square$

### C.3   Proof of the Theorem 3

**Theorem 3** (Restatement). *Under Assumption 2-5, the gap in performance between the policy induced by the learned reward function $r_{\hat{\theta}}$ and the ground truth optimal policy can be expressed as*

$$G(\pi^*, \hat{\pi}) \leq 2b_1\underline{\beta}\|\Delta\|_{\Sigma_D(\theta^*)+\lambda I_d}.$$

*Proof.* According to Lemma 1, under Assumption 2-5, for the MLE solution $r_{\hat{\theta}}$, we have

$$G(\pi^*, \hat{\pi}) = \mathbb{E}_{s\sim\rho}\mathbb{E}_{\tau\sim\pi^*}[r_{\theta^*}(\tau) - r_{\hat{\theta}}(\tau)] + \mathbb{E}_{s\sim\rho}\mathbb{E}_{\tau\sim\hat{\pi}}[r_{\hat{\theta}}(\tau) - r_{\theta^*}(\tau)].$$

Then, combined with Assumption 2 and 5, we can further obtain

$$\begin{aligned} G(\pi^*, \hat{\pi}) &= \mathbb{E}_{s\sim\rho}\mathbb{E}_{\tau\sim\pi^*}[r_{\theta^*}(\tau) - r_{\hat{\theta}}(\tau)] + \mathbb{E}_{s\sim\rho}\mathbb{E}_{\tau\sim\hat{\pi}}[r_{\hat{\theta}}(\tau) - r_{\theta^*}(\tau)] \\ &\leq 2b_1\|\Delta\|_2 \\ &\leq 2b_1\underline{\beta}\|\Delta\|_{\Sigma_D(\theta^*)+\lambda I_d}, \end{aligned}$$

The last inequality holds due to the compatibility of the norms, i.e., $\|v\|_2 \leq \underline{\beta}\|v\|_{\Sigma_D(\theta^*)+\lambda I_d}$ holds, for any $v \in \mathbb{R}^d$.

This completes the proof. $\qquad\square$

### C.4   Proof of the Theorem 4

**Theorem 4** (Restatement). *Under Assumption 2-5, the gap in performance between the policy induced by the learned reward function $r_{\hat{\theta}}$ and the ground truth optimal policy can be expressed as*

$$G(\pi^*, \hat{\pi}) \leq 2b_1\hat{\underline{\beta}}\|\Delta\|_{\Sigma_D(\hat{\theta})+\lambda I_d}.$$

*Proof.* According to Lemma 1, under Assumption 2-5, for the MLE solution $r_{\hat{\theta}}$, we have

$$G(\pi^*, \hat{\pi}) = \mathbb{E}_{s\sim\rho}\mathbb{E}_{\tau\sim\pi^*}[r_{\theta^*}(\tau) - r_{\hat{\theta}}(\tau)] + \mathbb{E}_{s\sim\rho}\mathbb{E}_{\tau\sim\hat{\pi}}[r_{\hat{\theta}}(\tau) - r_{\theta^*}(\tau)].$$

Then, combined with Assumption 2 and 5, we can further obtain

$$\begin{aligned} G(\pi^*, \hat{\pi}) &= \mathbb{E}_{s\sim\rho}\mathbb{E}_{\tau\sim\pi^*}[r_{\theta^*}(\tau) - r_{\hat{\theta}}(\tau)] + \mathbb{E}_{s\sim\rho}\mathbb{E}_{\tau\sim\hat{\pi}}[r_{\hat{\theta}}(\tau) - r_{\theta^*}(\tau)] \\ &\leq 2b_1\|\Delta\|_2 \\ &\leq 2b_1\hat{\underline{\beta}}\|\Delta\|_{\Sigma_D(\hat{\theta})+\lambda I_d}. \end{aligned}$$

The last inequality holds due to the compatibility of the norms, i.e., $\|v\|_2 \leq \hat{\underline{\beta}}\|v\|_{\Sigma_D(\hat{\theta})+\lambda I_d}$ holds, for any $v \in \mathbb{R}^d$.

This completes the proof. $\qquad\square$

