# OpenReview forum: "A Tighter Bound for Reward Learning in Reinforcement Learning from Human Feedback"
_TMLR — Accepted by TMLR_

### Review · Reviewer_WLtK · 2026-01-31

**Summary Of Contributions:**

The main contributions of the paper are the derivations of tighter bounds for reward learning in RLHF that are not dependent on constants such that the bounds approach zero as the dataset approaches infinite size. Another key difference from previous bounds is the proposed bounds are not restricted to the linear reward function class. The authors suggest $\underline{\alpha}$ which is the minimum eigenvalue of the empirical fisher information matrix $\Sigma_{D}$ is the most crucial parameter. They propose a constrained offline reward learning algorithm with a constraint on the empirical fisher matrix to keep the parameters in the confidence region of the dataset and extend it to online reward learning. A toy experiment was used to demonstrate improved reward learning accuracy of the online RLHF algorithm compared to uniform data sampling.

**Additional Comments:**

* I am wondering whether the bound for the linear case is guaranteed to be non-negative, since according to the equation between (22) and (23), you have a term $1/(1 - \beta^2\lambda)$.
* How are the proposed algorithms different from active preference learning where you sample data in the uncertain region and thereby increase confidence, e.g., [1]?

[1] [Bıyık, E., Talati, A., & Sadigh, D. (2022, March). Aprel: A library for active preference-based reward learning algorithms. In 2022 17th ACM/IEEE International Conference on Human-Robot Interaction (HRI) (pp. 613-617). IEEE.](https://arxiv.org/abs/2108.07259)

**Audience:**

Yes

**Audience Explanation:**

Understanding the theoretical behavior of RLHF and more efficient data sampling method is of wide interest.

**Claims And Evidence:**

Yes

**Claims Explanation:**

The main claim of the paper is a set of tighter bounds for RLHF. Compare to previously developed bounds, e.g., by Zhu et al, 2023 and Zhan et al, 2023a, the bounds are indeed tighter and approach zero as dataset increases. And these prior work are properly discussed and no significant prior work is missing as far as I can tell.

The toy experiment also clearly demonstrate the advantage of the proposed algorithm where targeted sampling improves accuracy over uniform sampling with small dataset size.

**Requested Changes:**

* I think the paper can do a much better job explaining the various quantities and constants to help readers understand the meaning of the proposed bounds more intuitively, in addition to just agreeing with the statements in the paper. Section 5.3 attempts to do this job and I think organizationally it's in an adequate spot in the paper. What I would like to see is the an explanation of most constants, e.g., $a_0, a_1, a_2, n^*$ if not all right after theorem 1.
* Another useful information intuitively is a comparison of (13) and (14), i.e., the bounds for $c_1=0$ (linear) and $c_1 > 0$ (non-linear). What are the main causes for the difference?

---

> ### Author Response · Authors · 2026-02-02
> **Response to Reviewer WLtK**
>
> We thank the reviewer for the insightful suggestions, which helped us better identify shortcomings and even errors in this paper. For the "Requested Changes":
> - Overall, the bounds depend on the parameters of the reward function class, i.e., $b_1,\, b_2,\,c_1,\,c_2,\,d$, the quality of the dataset, i.e., $\underline{\alpha}$, and the number of samples, i.e., $n$, while
> $a_2$ characterizes the part of the reward function class that captures nonlinearity.
>
> - Comparing (13) and (14), the linear-case bound can be viewed as a special case of the nonlinear one. Since $a_2 = 0$ for linear case,
> the bound in (13) can be further simplified to the form in (14). This relationship is analogous to a quadratic equation with a second-order coefficient.
> More clearly, the linear case bound can be written as
> $\frac{1}{\zeta(1-\frac{\lambda}{\lambda+\underline{\alpha}})}\sqrt{\frac{d+2\sqrt{\frac{d}{n^2}\log\frac{{2}}{\delta}}
> 		+2\log\frac{{2}}{\delta}}{n}}$.
>
> These changes are all highlighted in blue in the revised manuscript.
>
>
> As for "Additional Comments",
>
> - The initial manuscript was indeed confusing, as $\underline{\beta}$ itself depends on $\lambda$. In the revised version, we show this dependence; in fact, the bound should be expressed as ${\frac{1}{\zeta(1-\frac{\lambda}{\lambda+\underline{\alpha}})}} \sqrt{\frac{d+2\sqrt{\frac{d}{n^2}\log\frac{{2}}{\delta}}
> 		+2\log\frac{{2}}{\delta}}{n}}$. $\underline{\alpha}$ is a non-negative number, which is the smallest eigenvalue of $\Sigma_D(\theta^*)$. Then,
> $\underline{\alpha}=0$ means that $||\Delta||$ may tend to infinity in some dimension while $||\Delta||_{\Sigma_D(\theta^*)}$ is bounded, thus making the discussion of the bound meaningless. We have added these details in the revised manuscript.
> - One of the most intuitive differences is that APReL encodes the reward function in a linear form (see Eq. (1)). In addition, the problem setting is different. Considers only the case without expert demonstrations for initialization, APReL proposes multiple active query strategies generated from different perspectives, and our design can be viewed as one particular direction within this framework. From the perspective of the optimization objective, the most closely related approach is Volume Removal. However, their method selects queries based on uncertainty over the distribution, whereas ours is driven by uncertainty in parameter learning.

---

> > ### Comment · Reviewer_WLtK · 2026-02-04
> >
> > Thank the authors for their clarifications and updates. The new experiments are great!
> >
> > Just a few follow ups:
> > * You mean (13) is a reduction of (14) rather than the other way around? And the reduction is due to (15)? I think this second part is worthing pointing out as the reader wouldn't have read (15) when you make the statement.
> > * What's the difference between "uncertainty over the distribution" and "uncertainty in parameter learning"?

---

> ### Author Response · Authors · 2026-02-04
> **Response to Reviewer WLtK**
>
> Thank you for your comments.
> 1. We agree with your observation and have revised the wording to improve clarity. Comparing (13) and (14), both are derived from  (15). Specifically, for the linear case where $a_2 = 0$, (15) can be further simplified to (13).
> 2. In my understanding, the distribution refers to the representational capacity of the reward function, whereas parameter learning focuses on the distribution over parameters. For linear structures, the differences might only lie in the forms of expression and the mathematical tools employed. Intuitively, an equivalent bridge seems to be found, but due to my limited abilities and reading, there doesn't appear to be a rigorous theoretical proof of equivalence . However, for non-linear cases, addressing the problem from the perspective of representational capacity becomes significantly more challenging. For instance, as noted in Zhan et al. (2023a), the expressivity of the optimal reward function is required as a necessary bridge.

---

> > ### Comment · Reviewer_WLtK · 2026-02-06
> >
> > Thank the authors for their responses. All my questions are addressed.

---

> > > ### Author Response · Authors · 2026-03-06
> > > **Response to Reviewer WLtK**
> > >
> > > We appreciate the reviewer's insightful comments for improving this manuscript.

---

### Review · Reviewer_wocM · 2026-02-01

**Summary Of Contributions:**

This submission studies reward learning in RLHF from a parameter estimation perspective under general (nonlinear) function approximation. A key difference from prior work is that the proposed bounds avoid dependence on reward-class complexity measures and reduce reliance on inaccessible optimal parameters.

The main contributions are: (1) a tighter finite-sample estimation-error bound for reward parameter learning, along with a computable variant; (2) a connection between reward estimation error and downstream policy performance gap; and (3) algorithmic implications, including an offline RLHF approach with constraints and an online iterative RLHF procedure motivated by improving the informativeness of collected comparisons. Overall, the bound provides practical insight into how improving data “informativeness” can improve generalization.

A main weakness is the limited empirical validation: the numerical experiments are small-scale, and the paper would benefit from more extensive ablations and evaluations to support the practical claims.

**Audience:**

Yes

**Audience Explanation:**

Reward learning is central to RLHF, and this paper contributes to an active research direction on understanding when and why reward learning converges under nonlinear function approximation, as well as how dataset properties affect estimation and generalization. The analysis offers an informative viewpoint that may be useful both for theory-driven readers and for practitioners thinking about data collection and training stability.

**Broader Impact Concerns:**

I do not see major ethical issues of this paper, since it is primarily theoretical and methodological.

**Claims And Evidence:**

Yes

**Claims Explanation:**

The core claims are supported by a coherent set of theoretical results (Theorems 1–4) with clearly stated conditions and explicit bounds. In particular, the main bound is formulated in a way that removes explicit dependence on the unknown optimal parameter $\theta^*$, and the recommended procedures (offline constraints and online iterative data collection) follow plausibly from the theory.

That said, while the theoretical development is convincing under the paper’s assumptions, the practical and algorithmic implications would be more compelling with stronger empirical evidence and clearer implementation details for realistic settings (e.g., how the key quantities are estimated efficiently in high-dimensional models). Overall, the theory is well-supported, but the practicality claims would benefit from additional demonstrations.

**Requested Changes:**

The paper’s practical takeaway is the proposed algorithmic framework. I would expect at least one additional experiment beyond the small toy example (e.g., using a standard benchmark or a realistic preference-learning setup) to demonstrate scalability and practical benefit. In particular, it would strengthen the paper to report: (1) reward estimation error, (2) downstream policy performance, and (3) runtime/compute overhead, ideally with ablation studies that isolate the effect of the proposed data-collection or constraint mechanisms.

---

> ### Author Response · Authors · 2026-02-03
> **Response to Reviewer wocM**
>
> We thank the reviewer for the comments and have added an RL experiment on MuJoCo. We constructed $10,000$ preference samples based on the default $r^*$.
>
> (1) Based on this, we evaluate the quality of reward learning using some preference-learning metrics, including accuracy as well as the Pearson and Spearman correlation coefficients.
> Initially, we extracted 100 trajectories for training the reward function. Then, we collected an additional 50 trajectories using our proposed method and compared them with 50 randomly sampled trajectories. The result is:
> | Proxy reward model | Accuracy | Pearson coefficient | Spearman coefficient |
> |--------------------|----------|---------------------|----------------------|
> | $r_{\hat{\theta}^1}$ | 73.39% | 0.5666 | 0.6616 |
> | $r_{\hat{\theta}^2}$ | 77.43% | 0.6789 | 0.7348 |
> | $r^{\text{Alg3}}_{\hat{\theta}^2}$ | **96.43%** | **0.9906** | **0.9929** |
>
> (2) We evaluated these reward function on the downstream policy learning.  As shown in Figure 1 of the revised version, adding only 50 trajectories during the second online collection made the proxy reward model almost equivalent to the real reward.
>
> (3) The primary limitation of the algorithm stems from the eigenvalue computation. This point is discussed in the revised manuscript. In our experiment, on an Intel i9-14900HX CPU, collecting 50 trajectories in this environment requires about 6 hours.
> This value is related to our candidate set of 10,000 trajectories as well as to the eigenvalue solver used. Since we adopt a standard algorithm, there is room for further improvement.
>
> We sincerely appreciate the reviewer’s suggestions and the related metrics, which have helped us improve the manuscript.

---

### Review · Reviewer_S8EA · 2026-02-02

**Summary Of Contributions:**

This paper makes a strong theoretical contribution to RLHF by deriving a tighter parameter estimation error bound under general function approximation settings. It avoids dependence on unattainable optimal parameters and addresses previous limitations of uncomputable or vacuous bounds that rely on the complexity of the reward function class. Additionally, the authors propose offline and online RLHF algorithms inspired by their theoretical findings, with promising empirical illustrations in a toy example.

**Audience:**

Yes

**Audience Explanation:**

The paper introduces theoretical advances with practical implications in RLHF, aligning well with TMLR’s mission. It is likely to be of interest to a significant portion of the readership engaged in RLHF.

**Claims And Evidence:**

Yes

**Claims Explanation:**

The authors claim to provide a tighter error bound for reward learning in RLHF under general function approximation, without relying on reward function class complexity or unattainable optimal parameters. These claims are supported by two main theorems (Theorems 1-4), with clearly stated assumptions (Assumptions 2–5) and well-structured proof sketches in the main text, along with full proofs in the appendix.

**Requested Changes:**

1. The reference formatting appears to be not correct, particularly with respect to missing parentheses in citations. The authors should ensure correct usage of `\citet` and `\citep` throughout the manuscript.

2. Although the paper is primarily theoretical, the current experimental evaluation is limited to toy problems with low-dimensional trajectory spaces. To improve practical relevance and broaden the paper’s appeal, it would be beneficial to include experiments on standard RLHF benchmarks, such as Gridworlds or CartPole with trajectory preferences, as well as simple reward modeling tasks in robotic feedback simulations (e.g., MuJoCo) or even LLMs.

3. The tightness of the theoretical bounds and the empirical success of the proposed algorithms critically depend on the parameter $\alpha$ (the minimum eigenvalue of a data-dependent matrix). The paper would be strengthened by an empirical analysis of how $\alpha$ varies across datasets, along with a sensitivity study of performance with respect to $\alpha_{\text{target}}$ in the constrained optimization (Eq. 25). Additionally, approximate methods for estimating $\alpha$ in high-dimensional settings should be discussed.

4. Computing the minimum eigenvalue and solving the bi-level optimization in Algorithm 3 may be computationally expensive in high-dimensional regimes. The authors should discuss potential scaling strategies or approximations to improve efficiency.

---

> ### Author Response · Authors · 2026-02-03
> **Response to Reviewer S8EA**
>
> We sincerely thank the reviewer for the insightful comments, which have significantly improved the quality of the manuscript.
>
> 1. We have revised the formatting of the references to ensure correct presentation.
>
> 2. We believe the reviewer's suggestion is valuable, and we have added an RL implementation in MuJoCo. The specific details are provided below. The environment's default reward function is denoted as $r*$. A preference dataset is then constructed by sampling a set of state pairs and assigning preferences according to $r^*$.
> The reward function maps the current state, next state, and action to a proxy reward value, comprising a total of $84$ learnable parameters due to the omission of the bias term in the output layer.
>
> We constructed $10,000$ preference samples based on the $r^*$.
> Then, for reward learning, $100$ samples were initially selected to train the model $r_{\hat{\theta}^1}$, whose performance is evaluated using accuracy, Pearson, and Spearman correlation coefficients. After the initial training, we further collected $50$ trajectories through random sampling and Algorithm III to obtain the proxy reward models $r_{\hat{\theta}^2}$ and $r^{\text{Alg3}}_{\hat{\theta}^2}$, respectively.
> A comparative analysis of these models is provided in the following table.
> **Table. Evaluation metrics on the overall preference dataset.**
>
> | Proxy reward model | Accuracy | Pearson coefficient | Spearman coefficient |
> |--------------------|----------|---------------------|----------------------|
> | $r_{\hat{\theta}^1}$ | 73.39% | 0.5666 | 0.6616 |
> | $r_{\hat{\theta}^2}$ | 77.43% | 0.6789 | 0.7348 |
> | $r^{\text{Alg3}}_{\hat{\theta}^2}$ | **96.43%** | **0.9906** | **0.9929** |
>
> We adopt TD3 as the downstream reinforcement learning algorithm.
> Fig. 1 in the revised manuscript shows the performance of four different reward functions ($r^*$, $r_{\hat{\theta}^1}$, $r_{\hat{\theta}^2}$ and $r^{\text{Alg3}}_{\hat{\theta}^2}$) in downstream DRL tasks, where the y-axis represents the true and unnormalized reward.
> It can be seen that the proposed online collection algorithm improves data collection efficiency and achieves performance on downstream tasks that is closer to training with ground-truth reward compared to random sampling. We sincerely thank the reviewer for this comment. This makes the experimental evidence in the paper more comprehensive.
>
> 3. We acknowledge that computing $\alpha$ involves solving for the minimum eigenvalue, which constitutes a key computational bottleneck. Since $\alpha$ depends jointly on both the data and the model parameters, it is difficult to make direct comparisons of $\alpha$ across different datasets without also accounting for the associated parameter configurations. We have added a discussion of $\alpha$ in Section 5.3. We acknowledge that this bound is not universally applicable, but it applies to a broader range of settings than existing results.
> We have added a discussion of $\alpha_{target}$. In practice, it originates from the second iteration; therefore, one can first evaluate $\alpha$ on the full dataset under the current parameter configuration and then set
> $\alpha_{target}$ to half of this value (or a slightly smaller value).
>
> 4. We are not particularly concerned about the bilevel optimization problem, as small-scale experiments indicate that only one or two iterations are required. Our greater concern is the efficiency of online data collection, including the computation of the minimum eigenvalue. In the revised manuscript, we discuss this issue and suggest using a greedy algorithm for online trajectory collection, as in our RL experiments.
>
> Eigenvalue computation in high-dimensional settings remains a significant challenge. This is also the reason that limited the scale of our initial version experiment. We note that iterative methods, such as LOBPCG, can be employed to improve efficiency. However, to the best of our knowledge, computing eigenvalues of high-dimensional matrices is still an active and important area of research in applied mathematics. If the proposed approach is to be deployed in tne LLM problems, more efficient eigenvalue solvers may be required.
>
> The core contribution of this paper is the tighter bound. As shown in Zhu et al. (2023), even for linear problems, eigenvalue computation is an unavoidable component when performing online data collection. With increasing computational resources and capabilities, we will explore more effective approximation methods in the future.
>
> We sincerely appreciate the experimental environment suggested by the reviewer to help us improve the manuscript.

---

### Decision · Action_Editor_Sy3Z · 2026-03-04

**Recommendation:** Accept as is

**Audience:**

Yes

**Audience Explanation:**

This is a useful paper for the RLHF approach, which is widely used in today's LLM fine-tuning.

**Claims And Evidence:**

Yes

**Claims Explanation:**

This paper analyzes the parameter estimation problem for reward learning in RLHF under general function approximation. The claims in this paper are supported by rigorous proofs and numerical experiments.  Although the experiments are simple, but this is mainly a theoretic paper.